

# High Riverine CO₂ Outgassing affected by Land Cover Types in the Yellow River Source Region

Mingyang Tian[1], Xiankun Yang[2], Lishan Ran[3], Yuanrong Su[1], Lingyu Li[1],

Ruihong Yu[1], Haizhu Hu[1*], Xi Xi Lu[1,4*]

[1]Inner Mongolia key laboratory of river and lake ecology, School of ecology and environment, Inner Mongolia University, Hohhot, 010021, China

[2]School of Geographical Sciences of Guangzhou University, Guangzhou, 510006, China

[3]Department of Geography, The University of Hong Kong, Hong Kong, China

[4]Department of Geography, National University of Singapore, 117570, Singapore

✉ *Corresponding author*

Xi Xi Lu Tel.: +86-471-4991469, Fax: +86-471-4991436, e-mail: geoluxx@nus.edu.sg

Haizhu Hu Tel.: +86-471-4991469, Fax: +86-471-4991436, e-mail: huhaizhu@163.com





**Abstract:** Rivers connect the land and the oceans, acting as both active pipes and containers transporting carbon and other substances from terrestrial ecosystems to aquatic ecosystems. Meanwhile, rivers can release huge amounts of $CO_2$ to the atmosphere. However, estimates of global riverine $CO_2$ emissions remain greatly uncertain owing to the absence of a comprehensive spatially and temporally $CO_2$

emissions measurement, especially in river source regions. In this study, riverine partial pressure of $CO_2$ ($pCO_2$) and $CO_2$ efflux ($FCO_2$) in the Yellow River source region under different landcover types, including glaciers, permafrost, wetlands, and grasslands, were investigated in April, June, August, and October 2016. The relevant chemical parameters and environmental parameters, including pH, dissolved oxygen (DO), and dissolved organic carbon (DOC), were analyzed to explore the main control factors of

riverine $pCO_2$ and $FCO_2$. The results showed that the rivers in the Yellow River source region were a net $CO_2$ source, with the $pCO_2$ ranging from 181 to 2441 µatm and the $FCO_2$ from -221 to 6892 g C m$^{-2}$ yr$^{-1}$. Both the $pCO_2$ and $FCO_2$ showed strong spatial and temporal variations. The average $FCO_2$ in August was higher than that in other months, with the lowest in October. In alpine climates, low temperature conditions played a crucial role in limiting biological activity and reducing $CO_2$ emissions. The lowest

$FCO_2$ values (-221 g C m$^{-2}$ yr$^{-1}$) were observed in the glacier and permafrost regions. By integrating seasonal changes of water surface area, the total $CO_2$ efflux was estimated at 0.37±0.49 Tg C yr$^{-1}$, which is significantly higher than previous studies. Although it is still a small proportion of $CO_2$ emissions compared with the whole Yellow River Basin, but there is a huge carbon emissions potential. Since the permafrost in the source region of the Yellow River is rich in large amounts of ice and organic carbon,

the continuously increasing temperature due to global warming will accelerate not only the mobilization of organic carbon in permafrost, but also the degradation of organic carbon by soil microorganisms. As a consequence, huge amounts of $CO_2$ release from soils and rivers is anticipated.

**Key words:** $pCO_2$, $CO_2$ outgassing; glaciers; permafrost; wetland; grassland; Yellow River source region



## 1. Introduction

Rivers connect the land and the oceans, acting as both pipes and containers transporting carbon and other substances from terrestrial ecosystems to aquatic ecosystems. At the same time, they receive organic or inorganic carbon from the terrestrial carbon pool, degrade and ultimately release as $CO_2$ or buried in the riverine sediments. The existing studies on riverine $CO_2$ evasion mainly focuses on the spatial and temporal dynamics of partial pressure of $CO_2$ ($pCO_2$) and $CO_2$ efflux ($FCO_2$). (Cole et al.,2001;

Aufdenkampe et al., 2011; Raymond et al., 2013; Abril et al., 2014). Many researchers believe that river water $CO_2$ is mainly derived from the respiration of terrestrial ecosystems and the decomposition of organic matter in river waters, but the source and impact mechanism of $CO_2$ evolution from rivers is still not clear (Raymond et al., 2013; Hotchkiss et al., 2015; Schelker et al., 2016; Ran et al., 2017). Therefore, in order to accurately estimate the riverine $CO_2$ outgassing and in-deep understanding its control

mechanism, more research within the river in particular climates (i.e., alpine climate) and special regions (i.e., headwater region or intermitted rivers) are needed. It will provide a better understanding of regional or global carbon balance processes and future climate change trends.

With respect to global-scale $CO_2$ outgassing, available estimates are characterized by great uncertainty.

For example, recent $CO_2$ outgassing fluxes from global rivers and streams combined range from 1.8 to 3.2 P g C yr$^{-1}$ (Raymond et al., 2013; Drake et al., 2017), which are significantly higher than earlier estimate by Cole et al. (2007) (i.e., 0.75 P g C yr$^{-1}$ based on the data of 80 major rivers in the world). A major reason for the huge range is because of the absence of a global $CO_2$ outgassing database which includes $CO_2$ emissions measurement over different rivers and under different climate and land cover

types (Raymond et al., 2013; Cole et al., 2007; Aufdenkampe et al., 2011; Drake et al., 2017). Thus, more field measurements based on global river systems are strongly needed to increase the accuracy of the



estimates.

However, there are limited studied on $CO_2$ effluxes of rivers in extreme geographical and climatic

conditions, such as alpine rivers (Wu et al., 2008; Zhang et al., 2013). Crawford et al. (2013) investigated

the riverine $CO_2$ outgassing in the Alaska region and explored its temporal and spatial changes by

connecting it to land use types. Crawford et al. (2015) further studied carbon emissions from the rivers

and lakes in alpine areas around Estes Park in the United States and found the average $pCO_2$ was only

417 µatm. They concluded that the high altitude and low vegetation coverage are the primary factors

limiting $CO_2$ outgassing. Weyhenmeyer et al. (2015) collected data from 5,118 alpine lakes and

concluded that the production of $CO_2$ in the lake was usually half of the $CO_2$ emissions and most of the

emitted $CO_2$ derived from dissolved inorganic carbon (DIC). Humborg et al. (2010) surveyed rivers in

central and northern Sweden and concluded that the average $pCO_2$ was 1445 µatm and the average $FCO_2$

value was 3033 g C m$^{-2}$ yr$^{-1}$. A comprehensive analysis indicated that groundwater and respiration of

soil maintained the riverine $CO_2$ excess with the consumption of terrestrial organic matter as the major

source of riverine $CO_2$. Overall, compared with the temperate and tropical rivers, riverine $CO_2$ outgassing

under the alpine climate is at a relatively low level. It is mainly due to the cold climate with low

temperature and high altitude that limit riverine $CO_2$ emissions, and the underlying control mechanisms

are not the same as these in temperate and tropical climates (Peter et al., 2014).


The riverine $CO_2$ emissions in the Yellow River Basin has been studied and some preliminary results

have been reported. Su et al. (2005) reported that the $pCO_2$ value of the mainstream was between 1100

and 1700 µatm, which were in the intermediate-low level of the world rivers. The main controlling factor

was its carbonate system. Zhang et al. (2008) measured $pCO_2$ of 1570 µatm at Lijin Hydrological Station

on the lower Yellow River during sediment regulation period (June–July), which was significantly higher

than in the other periods. Zhang et al. (2009) measured the $F\mathrm{CO}_2$ of the Yellow River and concluded that

the Yellow River water was a source of atmospheric $CO_2$ during the autumn. The amount was about

0.0174 Tg C, and the flux was similar to that of the Ottawa River but far less than that of the Amazon.

Ran et al. (2015b) estimated that the annual $CO_2$ emissions of the whole Yellow River at 7.9 Tg C, which

is close to the basin-wide carbon deposition of 8.7 Tg C while larger than the amount of marine import

(i.e., 6 Tg C).

These studies on $CO_2$ emissions from the Yellow River were mainly confined to its middle and lower

reaches and the estuary. In contrast, there are few studies on the upper reaches, especially the source

region on the Tibetan Plateau. The Yellow River source region is located in the alpine zone with the

Yellow River mainstream and its tributaries flowing through a variety of land cover types, including

grassland, wetland, glacier, and permafrost. Affected by increasing temperature as a result of global

warming, the alpine rivers in this region have become hot spots of riverine carbon cycle studies and

warrant a thorough understanding of their implications in the context of global climate change (Ulseth et

al., 2018; Peter et al., 2014; Hood et al., 2015). In particular, although Ran et al., (2015a,b) used compiled

water chemistry data to estimate $p\mathrm{CO}_2$ and $F\mathrm{CO}_2$, there are no field-based direct measurements of $CO_2$

emissions from these alpine rivers.

In order to accurately determine the intensity of riverine $CO_2$ outgassing and fully understand the

underlying control mechanisms in the alpine climate region, we conducted in situ measurements of $CO_2$

emissions from the alpine rivers under different land cover types, including grassland, peatland, glacier,

and permafrost, in the Yellow River source region. Here we aim to address three questions regarding $CO_2$

emissions in the Yellow River source region: (1) the spatiotemporal patterns of $CO_2$ emissions under different land cover types; (2) the magnitudes of stream $CO_2$ emissions; and (3) the source of riverine

$CO_2$ in this alpine river system. Answers to these questions will lead to a better understanding of riverine carbon export and $CO_2$ emissions, especially for alpine rivers, which will help refine the global estimation of riverine $FCO_2$.

## 2. Materials and methods

### 2.1 Site description

The Yellow River originates from the north part of the Bayanhar Mountains in the Tibetan Plateau, then

flows through the Loess Plateau and the North China Plain, and eventually empties into the Bohai Sea.

Generally, the drainage basin above the Toudaoguai hydrological station is called the upper reach and the

region above the Tangnaihai hydrological station is known as the Yellow River source region (Figure 1).


The study area is situated from 32°3'N 95°5'E to 36°1'N 103°3'E (Figure 1). In the Yellow River source

region, most of the rivers flow through the Tibetan Plateau at an altitude of 3000–4000 m with

meandering river channels. The study area is about $1.32 \times 10^5$ km$^2$, accounting for about 17.6 % of the

Yellow River basin. The Yellow River source region is located in an alpine zone, which is a typical

plateau continental climate, mainly affected by plateau monsoon. The northern part belongs to the semi-

arid climate zone, while the middle and southern part is in the humid and sub-humid climate zone (Yang

et al.,1991).

The annual mean precipitation is 486 mm, which is the dominant factor of runoff (Sun et al., 2009),

accounting for approximately 95.9% of the total runoff in the source area (Liu et al., 2005). The annual

evaporation varies from 800 to 1200 mm. Although the area of source region accounts for only 17.6% of the whole basin, it supplies over 33% water of the Yellow River, providing an important water resource for both middle and lower reaches of the Yellow River (Sun et al., 2009). In recent decades, although the precipitation has slightly increased (Chang et al., 2007), the water discharge in the middle and lower

reaches has decreased significantly, which has aggravated the water shortage of the downstream region, especially in the non-flood season (Zhang et al., 2012).

### 2.2 Fieldwork and laboratory analysis

In this study, four rounds of field work in April, June, August, and October 2016 were conducted. The

riverine $p$CO$_2$ and related environmental factors, including water temperature, pH, dissolved oxygen (DO), were monitored in the field under different land cover types in the Yellow River source region. In total, there are 36 sampling points (Figure 1) within the study area and they can be categorized on the basis of the complexity of river network structure and the land cover types (i.e., glacier, permafrost, wetland, and grassland) (Table 1). In addition, three groundwater samples in grassland covered area were

taken to determine its $p$CO$_2$. The temperature, pH, and DO were measured by using a Multi 3420 analyzer (WTW, Germany) with the accuracies of ±0.2 °C, ±0.004, and ±1.5%, respectively. Before the measurement, the pH probe was calibrated by three pH buffers (e.g., pH 4.01, pH 7.00, and pH 10.01, respectively).

The prior study suggested that, when the pH ranges from 7 to 10, HCO$_3^-$ represents 96% of alkalinity, alkalinity can be used to calculate DIC (Hunt et al., 2011). Alkalinity was determined by on-site titration. The collected water sample was subjected to low-pressure suction filtration through a glass fiber filter (Whatman GF/F) with a pore diameter of 0.7 μm. The fiber filter was pre-fired in a muffle furnace at

450 °C. For each water sample, the alkalinity was titrated with 0.1 mol $L^{-1}$ HCl within 12 hours after

sampling. Each titration was repeated three times to assure the analytical error below 3%. The Methyl

orange indicator was used to determine the endpoint of the reaction at pH=4.5. Another 100 mL of the

filtered water sample was transferred into the specific bottle, added with nitric acid, and preserved in

refrigerator at 4 °C condition for dissolved organic carbon (DOC) measurement in laboratory. DOC was

analyzed with the Vario total nitrogen/organic carbon analyzer (Elementar, German), which has a

precision less than 3%.

### 2.3 Calculation of $CO_2$ emission

In this study, $FCO_2$ was measured by the floating chamber method (Ran et al.,2017) connected with a

Li-7000 $CO_2$/$H_2O$ analyzer (Li-Cor, Inc, USA). The Li-7000 instrument was calibrated with standard

$CO_2$ gases of 500 ppm and 2000 ppm before each measurement.

The volume of rectangular floating chamber is 17.8 L and the covered water area is 0.09 $m^2$. The chamber

walls were lowered 3 cm into the water and mounted with plastic foams that had streamlined ends to

limit artificial disruptions to near-surface turbulence. The chamber is covered with tin foil to reduce the

influence of sun light's heating. The temperature inside the chamber was measured with a waterproof

thermometer. At the beginning of each experiment, the chamber was placed in the air near the monitoring

point and the air inside the chamber was continuously circulated in a closed loop that was connected to

an infrared Li-7000 gas analyzer through rubber-polymer tubes for $CO_2$ analysis. The instrument

automatically records the air $CO_2$ concentration and ambient atmospheric pressure. When the chamber

was placed on the water surface, the analyzer recorded the $CO_2$ concentration every 2 seconds, and each

measurement lasted for 6–10 mins. In large rivers with relatively favorable flow conditions, we fixed the

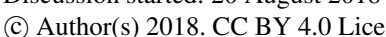

chamber on a small rubber boat and drifted along the water to measure the $F\mathrm{CO_2}$. In contrast, we used the static chamber method to measurement the $F\mathrm{CO_2}$ in the small rivers or streams which could lead an overestimate of $\mathrm{CO_2}$ evasion to some extent. (Lorke et al., 2015).


The $\mathrm{CO_2}$ flux from water is calculated using the following equation (Frankignoulle et al., 1988):

$$F\mathrm{CO_2}=1000\times(dp\mathrm{CO_2}/dt)\,(V/RTS) \qquad (1)$$

where, $dp\mathrm{CO_2}/dt$ is the slope of $\mathrm{CO_2}$ change within the chamber (Pa d$^{-1}$; converted from µatm min$^{-1}$), $V$ is the chamber volume (17.8 L), $R$ is the gas constant, $T$ is chamber temperature (K), and $S$ is the area of

the chamber covering the water surface (0.09 m$^2$).

Conventionally, $F\mathrm{CO_2}$ can also be estimated from the following equation.

$$F\mathrm{CO_2}=k\cdot\mathrm{K_H}\cdot\Delta p\mathrm{CO_2} \qquad (2)$$

Where, $k$ is gas transfer velocity (cm h$^{-1}$), $\mathrm{K_H}$ is the Henry's constant for $\mathrm{CO_2}$ at a given temperature, the $F\mathrm{CO_2}$ is the in situ measured riverine $\mathrm{CO_2}$ efflux, and the $\Delta p\mathrm{CO_2}$ is the difference between the surface

water and the atmosphere. Using the field-measured $p\mathrm{CO_2}$ in surface water and air, $k$ can be computed by rearranging Equation (2). In order to facility compare our $k$ with the results of other studies, we standardized it to a Schmidt number of 600 ($k_{600}$) by assigning the Schmidt number exponent value of 0.5 (Jähne et al., 1987).

Surface water $p\mathrm{CO_2}$ was calculated using a headspace equilibrium method (Ran et al., 2017). By using an 1100 mL conical flask, 800 mL of water were collected in the depth of 10 cm below the water surface and the remaining volume of 300 mL was filled with ambient air. The flask was immediately closed with a lid and vigorously shaken for 1 min to equilibrate the gas in water and air. The equilibrated gas was

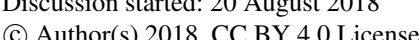



automatically injected into the calibrated Li-7000 gas analyzer. The measurements at each site were

repeated 3 times and the average was calculated (analytical error below 3%). Surface water $pCO_2$ was

calculated based on the equations from Dickson et al. (2007):

$$pCO_2^{water,i} \approx pCO_2^{headspace,f} + \frac{Vh}{Vw}(pCO_2^{headspace,f} - pCO_2^{headspace,i})/(K_0RT)$$

Where, the superscripts $i$ and $f$ represent initial and final $pCO_2$ (µatm), $Vh$ and $Vw$ are the headspace

volume and water volume, respectively, $K_0$ is the solubility of $CO_2$ in water calculated on the basis of

solubility constants for $CO_2$ from *Weiss* (1974), $R$ is the universal gas constant (8.314 J mol$^{-1}$ K$^{-1}$), and $T$

is the water temperature on Kelvin scale (K). Temperature in the flask after equilibration was measured

to correct for changes in temperature compared to in-situ water. The initial $pCO_2$ was taken as the $CO_2$

concentration in ambient air before the headspace equilibration measurement.

**3.Results**

**3.1 Characteristics of hydro-chemical variables**

Water temperature (Tw) varied from 0.1 to 27.7 °C with an average of 11.9±5.7 °C. Average Tw in June

(15.1±3.5 °C) and August (17.0±5.4 °C) is significantly higher than that in April (8.4±3.8 °C) and

October (7.3±2.4 °C). Seasonal Tw difference was more significant at the wetland (14.4±6.4 °C) and

grassland (12.5±5.4 °C) sites than that in the glacier (7.5±4.1 °C) and permafrost (10.0±4.0 °C) sites.

These results were expected as the water temperature depends mainly on the air temperature. Spatial

variability of the air temperature was consistent with that of the water temperature at almost all the sites,

although in some case it was as high as 33 °C. The annual average air temperature in the study area

was 16.7±6.3 °C.


Water pH ranged from 7.0 to 9.0 with an average of 7.9±0.6 (Table 1). Mean pH based on all the stream





samples was 8.3±0.4, 8.6±0.4, 7.2±0.2, and 7.5±0.4 in April, June, August, and October, respectively. A slight decreasing trend is observed with the different land cover types in the order of permafrost > glaciers > grassland > wetland, with the average pH value at 8.1±0.9, 7.93±0.55, 7.85±0.59, and 7.7±0.5,

respectively (Table 1). Alkalinity ranged from 600 to 7600 μmol L$^{-1}$ with an average 2871±1381 μmol L$^{-1}$ (Table 1). Alkalinity was higher in the cold months (3378 μmol L$^{-1}$ in April and 2941 μmol L$^{-1}$ in October) than in the warm months (2644 μmol L$^{-1}$ in June and 2326 μmol L$^{-1}$ in August). Alkalinity of the river water in glaciers covered area showed consistently the lowest level throughout the year (Table 1), due to the low coverage of carbonate rocks.


DO values ranged from 2.7 mg L$^{-1}$ to 12.1 mg L$^{-1}$ and the basin-wide mean DO was 7.8±0.6 mg L$^{-1}$ in April, 7.1±1.4 mg L$^{-1}$ in June, 6.7±0.7 mg L$^{-1}$ in August and 7.7±0.7 mg L$^{-1}$ in October, respectively (Table 1). For the land cover types, the highest level of DO were in the glacier, with the annual average of 7.6±0.8 mg L$^{-1}$, followed by the permafrost with 7.4±1.4 mg L$^{-1}$, the grassland with 7.3±0.9 mg L$^{-1}$,

and peatland with 7.2±1.1 mg L$^{-1}$, respectively (Table 1).

DOC ranged from 0.2 to 12.2 mg L$^{-1}$ with an average of 4.7±2.7 mg L$^{-1}$ (Table 1). DOC exhibited strong seasonality across the rivers. The highest DOC concentration occurred in April (5.0±1.6 mg L$^{-1}$), followed by in August (4.9±3.6 mg L$^{-1}$) and June (4.7±2.9 mg L$^{-1}$), and the lowest was found in October

(4.0±2.2 mg L$^{-1}$). For the land cover types, the highest level of DOC was in the peatland covered area, with the annual average of 5.1±3.7 mg L$^{-1}$, followed by the permafrost with 4.9±2.4 mg L$^{-1}$, the grassland with 4.6±2.3 mg L$^{-1}$, and the glaciers with 3.4±1.1 mg L$^{-1}$, respectively (Table 1).

We used the measured flow velocity and channel slope to predict the $k_{600}$ based on the Model 5 presented





by Raymond et al. (2012). The computed $k_{600}$ showed strong statistically significant but weak agreement

with the model results (Figure 2a). Given the chamber's dampening effect of wind (Matthews et al.,

2003), there was not any statistically significant relationship between wind and $k_{600}$ for streams. Instead,

flow velocity is a relatively good predictor variable of $k_{600}$ and can approximately explain 15% of its

variability (Figure 2b). Although we deployed the floating chamber very carefully, but the whole

statistical analysis could not reflect the multi interaction of variety environment factors beyond different

land cover types through our 36 sampling sites. Additionally, in some sampling points, the Model 5

overestimated some $k_{600}$ values especially in some mountainous rivers, mainly due to the water

temperature played a crucial role in limiting $CO_2$ transfer between the air-water interface in the plateau

region although the higher channel slope supported enough condition for water turbulence (Battin et al.,

255    2008).

### 3.2 Spatial and temporal variations of $p$CO$_2$

The $p$CO$_2$ ranged from 181 to 2441 µatm with an average of 774±377 µatm, nearly twofold the ambient

air $p$CO$_2$. To better illustrate the spatial variability $p$CO$_2$, Figure 3a, 4a, and 3c showed its changes with

land cover types. The highest average $p$CO$_2$ value appeared in the peatland covered area (937±466 µatm),

followed by grassland (818±394µatm), glacier (645±253 µatm), and the permafrost (600±212 µatm).

The $p$CO$_2$ value showed different temporal variation characteristics for the four land cover types (Figure

3a, 4a, and 4c). In grassland covered area, the average river $p$CO$_2$ value in April, June, August, and

October was 836±258 µatm, 609±297 µatm, 1086±551 µatm, and 734±253 µatm, respectively. In

peatland covered area, the average river $p$CO$_2$ value of April, June, August and October was 875±436

µatm, 792±436µatm, 1156±630 µatm and 926±285 µatm, respectively. The water $p$CO$_2$ in these two land

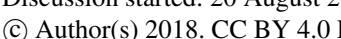


cover types showed the same temporal variation pattern, with the highest $pCO_2$ occurred in August and

the lowest in June for all the land cover types.


Unlike in the peatland and grassland regions, the riverine $pCO_2$ in the glacier and permafrost regions

showed relatively small variations, but still had the similar seasonal variation trends. In the glacier

covered area, the average river $pCO_2$ value of April, June, August and October was 635±122 µatm,

506±31 µatm, 738±449 µatm, and 632±132 µatm respectively. In the permafrost covered area, the

average river $pCO_2$ value of April, June, August, and October was 465±216 µatm, 586±227 µatm, 591±74

µatm, and 756±231 µatm, respectively.

### 3.3 Spatial and temporal variations of $FCO_2$

$CO_2$ emissions exhibited significant spatial and seasonal variations among the 36 stream sites (Table 1,

Figure 3b, 4b, and 4d). The $CO_2$ effluxes ranged from -221 to 1469 g C m$^{-2}$ yr$^{-1}$ in April, -144 to 6892 g

C m$^{-2}$ yr$^{-1}$ in August, and -34 to 2321 g C m$^{-2}$ yr$^{-1}$ in October. While the highest $FCO_2$ was measured at

the wetland covered sites (Site Pt 3 in August, 6892 g C m$^{-2}$ yr$^{-1}$), the lowest $FCO_2$ was observed at

permafrost covered sites (Site Pm 3 in April, -221 g C m$^{-2}$ yr$^{-1}$) (Table 1). The averaged $FCO_2$ of all sites

was 479±436, 261±205, 873±1220, and 714±633 g C m$^{-2}$ yr$^{-1}$ in April, June, August, and October,

respectively. Clearly, rivers in the Yellow River source region were net carbon sources for the atmosphere,

despite the great $FCO_2$ variations over space and time. Grouped by land cover types, the mean $CO_2$ efflux

shows a significant decreasing trend from wetland (767±1644 g C m$^{-2}$ yr$^{-1}$), grassland (679±610 g C m$^{-}$

$^2$ yr$^{-1}$), glacier (508±588 g C m$^{-2}$ yr$^{-1}$) to permafrost (302±±349 g C m$^{-2}$ yr$^{-1}$). Because the intensity of

$FCO_2$ depends on $pCO_2$ in stream water, the $FCO_2$ showed a similar spatial and temporal pattern to the

$pCO_2$, although the highest and lowest $pCO_2$ and $FCO_2$ value were not found at the same sampling sites.





## 4. Discussion

### 4.1 Impact of land cover types on riverine $pCO_2$ and $CO_2$ outgassing

Among all land cover types, the lowest $FCO_2$ appeared in the permafrost covered region, with the annual

average $FCO_2$ of 302±349 g C m$^{-2}$ yr$^{-1}$. It is well known that the riverine $CO_2$ is derived from land

(Dinsmore and Billett., 2013; Hope et al., 2004), and the area covered by permafrost has a high density

of organic carbon in soil (Zeng et al., 2004), these soil organic carbons can support large quantities of

DOC to the rivers and cause enormous riverine $CO_2$ outgassing. The correlation analysis between various

hydro-chemical parameters and $pCO_2$ in the permafrost region showed that alkalinity, DO were not highly

correlated with $pCO_2$ and pH, DOC indeed had a close relation with $pCO_2$ (Figure 5). The negative

relationship between $pCO_2$ and pH is explained as dissolved $CO_2$ acts as an acid in water (Stumm and

Morgan., 1996), and in poorly buffered systems, $CO_2$ can be a strong control on the stream pH (Neal et

al., 1998; Waldron et al., 2007). This positive correlation between DOC and $pCO_2$ suggests the terrestrial

related DOC might support partial riverine $CO_2$ concentration (Liu et al., 2016). This indicates that DOC

is one of the important sources of permafrost river $CO_2$. The DOC concentration in the rivers in the

permafrost covered area is relatively high, averaging at 5.0±2.4 mg L$^{-1}$, which exceeded 3.6±1.1 mg L$^{-1}$

in glacier areas and 4.6±2.3 mg L$^{-1}$ in grasslands, but was close to 5.1±3.7 mg L$^{-1}$ in peatlands, and

sometimes even exceeded the DOC concentration of rivers in the peatland covered region. Additionally,

the average alkalinity concentration in that region is the highest among four types. However, the $pCO_2$

and $FCO_2$ value in this region were always the lowest during the four campaigns. This was due to the

area covered by permafrost usually with the highest elevation and the lowest average temperature among

the four types of land cover with the average water temperature around 9.99 ℃. These conditions limited

the soil respiration and riverine organic matter degradation (Battin et al., 2008). Additionally, in terms of

gas diffusion, although there is sufficient dissolved $CO_2$ in the river water, it is not easy for $CO_2$ emission

from rivers to the atmosphere in the condition of low temperature and low flow velocity (average: $0.8\pm0.5$ m s$^{-1}$) (Alin et al., 2014). In summary, the lower temperature is the main cause of high riverine DOC concentrations and low $CO_2$ outgassing rate in the permafrost covered region.

The glaciers covered region has the similar temperatures and elevations to the permafrost, thus its $pCO_2$

and $FCO_2$ values were also lower in the permafrost covered region, with the average value only at $657\pm240$ g C m$^{-2}$ yr$^{-1}$. This is probably because all the sampling sites are located on the 1–2 order streams characterized by strong hydrologic connection with the terrestrial landscape (Sorribas et al., 2017; Smits et al., 2017), and the surrounded environment lack of exogenous terrestrial carbon support. For the glacier area, only DOC was related to $pCO_2$ (Figure 6d, r$^2$=0.56, p < 0.001). The sampling points under the

glaciers are mainly located around the Aemye Ma-chhen Range. Some glacial sampling sites all have some ice and snow melting water supply. The glacial-covered river water has the lowest DOC concentration with a lowest value of four land cover types ($3.6\pm1.1$ mg L$^{-1}$). The area around the Aemye Ma-chhen Range without enough vegetation coverage because of the harsh environment of high elevation and low annual average water temperature, limiting the DOC source. Poor soil, short water retention, and

low precipitation are the main reason of the low vegetation coverage in this region (Lu et al., 2001). The river near the sampling sites of the snow mountain has been cut deep into the B horizon of soils as a result of glacial erosion and retreat. Almost all glacial sampling sites are covered with gravel, limiting the supply of terrestrial organic carbon to river carbon pools. As a result, the measured DOC concentrations in most glacial areas were very low. In glacial rivers, if there is no external supply of DOC,

the whole water DOC contribution only amounted to 0.34 µmol L$^{-1}$ $CO_2$ gain. This highlight that the $CO_2$ produced by DOC degradation in the glacial river cannot maintain such a high $CO_2$ outgassing rate.




Although there is low content of carbon in ice and snow of that region (Wu et al., 2008), the meltwater of ice and snow continues to erode the surrounding bedrock during long-distance transport, resulting in more limestone in the rivers. Previous studies have shown that glaciers contain large amounts of $CO_2$

(Meese et al., 1997) and DOC (Hood et al., 2009; Singer et al., 2012), which are important sources of $CO_2$ in glacier rivers. Our observations found that with the increasing distance from the Aemye Machhen Range, the riverine $pCO_2$ exhibited a decrease trend, which could be explained by the dilution effect of water snow-melting water $pCO_2$. Therefore, the $CO_2$ in the region is highly likely to be explained by the $CO_2$ storage from glacier. As the glaciers melt, $CO_2$ in the glaciers was brought into the river.


The river $FCO_2$ is the highest in the peatland coverage area among the 4 studied types. The relation showed that only pH has a negative linear relationship and alkalinity have a weak linear relationship with the $pCO_2$ (Figure 7). In the peatland rivers, the terrestrial-related organic carbon is an important source of riverine $CO_2$ (Abril et al., 2014; Müller et al., 2015; Billett et al., 2015, Hu et al.,2015). There are many

sources of DOC in the peatland. First, the soil in the wetland ecosystem is rich in peat soil. The amount of peat stock in Zoige Peatland is estimated to be 1.9 billion tons, which accounts for about 40% of China's marsh wetland carbon storage (Wang et al., 2012). These carbon supplies to river carbon pools are an important driver for the high $FCO_2$ in the wetlands rivers. On the other hand, soil pore water enriched with high concentrations of dissolved $CO_2$ continues to enter river waters, and it also provides

sufficient carbon sources for rivers (Butman et al., 2011). In addition, the vegetation in the peatland region can also import large amounts of $CO_2$ into the river water through two other mechanisms. First, vegetation litter and root exudates release unstable organic carbon into the water. These organic carbons are being further decomposed and served as a carbon source for heterotrophic microorganisms. During this process, heterotrophic organisms release $CO_2$ into water (Abril et al., 2014). On the other hand, the

respiration of plant roots and soil microorganisms that are submerged in wetland soils releases $CO_2$ directly into the water (Abril et al., 2014). The combined effects of these factors have resulted in rivers with high DOC and high $FCO_2$ value in wetlands.

The average $FCO_2$ in the grassland-covered rivers of 818±394 g C m$^{-2}$ yr$^{-1}$ is at a moderate level, below

the wetland $FCO_2$ but significantly higher than the riverine $FCO_2$ in the glaciers and permafrost covered regions. Correlation analysis between water chemistry parameters and riverine $pCO_2$ in this area showed that both pH and DOC had weak correlation with $pCO_2$ (Figure 8). This also shows that the pH of river water in the area is partial affected by the $CO_2$ concentration in water. Due to the temperate environment, grassland is the most human-affected area in the study area, mainly in the form of grazing. As a result,

beside the DOC derived from the physical erosion, the pollutants produced by grazing are also important sources of riverine DOC. The average $pCO_2$ in peatland is 15% higher, but the average DOC concentration in wetlands is 11% higher than that in grassland, and the alkalinity in grassland is 46% higher than that in wetlands. Therefore, DIC is an important source of riverine $CO_2$ in grasslands. While stream DIC source are highly variable across space and time (Smits et al., 2017), the DIC mainly

originated from groundwater (Marx et al., 2017). Although groundwater is participated in the carbon cycle of the river in the entire study area, it is higher in the grassland than in other regions, indicating that the supplemental effect of groundwater on the river $CO_2$ in the grassland is the biggest. We also take three groundwater samples in grassland covered area. The average $pCO_2$ of groundwater samples in grassland-covered areas is 1976 µatm, which is 2.5 times the average value of the water in the Yellow

River source region. Therefore, in the grass-covered areas, the $CO_2$ excess in the rivers is maintained by both the terrestrial vegetation organic carbon and the inorganic carbon in the groundwater.

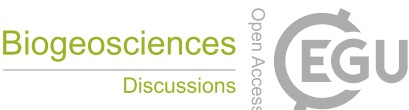


### 4.2 Significance and implications for riverine carbon budgets

The annual average $p$CO$_2$ is 771±380 µatm and $F$CO$_2$ is 590±766 g C m$^{-2}$ yr$^{-1}$ in the Yellow River source

385 region. In the whole Yellow basin, Ran et al. (2015a, b) estimated the a significantly lower $p$CO$_2$ value

of 241±79 µatm and the areal CO$_2$ efflux of this region is -221±112 g C m$^2$ yr$^{-1}$, indicative of a strong

carbon uptake from the atmosphere. Combining the water surface area of wet season (122 days and the

area of 770 km$^2$) and dry season (243 days and the area of 560 km$^2$), we estimated total CO$_2$ efflux from

the Yellow River source region at about 0.37±0.49 Tg C yr$^{-1}$, suggesting a net carbon source for the

390 atmosphere. This efflux contrasts with the earlier estimate by Ran et al. (2015b) which reported a carbon

sink of -0.168±0.084 Tg C yr$^{-1}$.

Unlike our systematic sampling within the Yellow River source region, Ran et al. (2015b) estimated the

riverine CO$_2$ outgassing of the Yellow River source region by only using sampling results at five sampling

395 sites. There are a number of reasons for the huge CO$_2$ efflux difference. Firstly, the sampling by Ran et

al. (2015b) was confined to the mainstream channel of the Yellow River and its major tributaries, which

may have underestimated riverine CO$_2$ emissions in lower-order streams. For example, our study on the

rivers of the Zoige peatland indicated that the higher $F$CO$_2$ was observed at the lower-order headstream

tributaries (i.e., 767±1144 g C m$^{-2}$ yr$^{-1}$) instead of the mainstream (i.e., 351±306 g C m$^{-2}$ yr$^{-1}$). This is

400 because the soil carbon-rich peatland around the rivers can be rapidly transported into the river network

by strong physical erosion. However, with increasing flow discharge and enhanced erosion, the river

channels are heavily cut into the bedrock, mobilization the subsurface soils with less soil carbon content

has likely caused the dilution effect of $p$CO$_2$ in the mainstream (Crawford et al., 2013). Another reason

is that the number of sampling points limited the accuracy of CO$_2$ emissions of relatively large watershed,

405 especially in the alpine area and intermitted rivers. This is due to the fact that in the rivers of the source

area, the high-concentration dissolved $CO_2$ groundwater is an important source of river $CO_2$, two groundwater samples collected in the grassland covered area showed an average $pCO_2$ of 1976 μatm, 2.5 times larger than that in the river (771±380 μatm). The $CO_2$ which originates from groundwater can be quickly released to the atmosphere within a short distance (Hotchkiss et al., 2015). In the case of point

deployment, this process is difficult to monitor without the high-density sampling points arrange. Therefore, it is easy to neglect this part of $CO_2$ by representing the entire basin with relatively few sampling points.

The area of the Yellow River source region account for about 17.6% of the whole Yellow River basin and

support around 4% of the total $CO_2$ efflux. Although it is still a small proportion of $CO_2$ emissions compared with the whole Yellow River Basin, but there is a huge carbon emissions potential. Since the permafrost in the source region of the Yellow River is rich in large amounts of ice and organic carbon, the continuously increasing temperature due to global warming will accelerate not only the mobilization of organic carbon in permafrost, but also the degradation of organic carbon by soil microorganisms. As

a consequence, huge amounts of $CO_2$ release from soils is anticipated and relevant studies will be needed to comprehensively understand the implications of changes in riverine carbon fluxes.

Although we have made some improvements to evaluate the riverine $CO_2$ emission in the Yellow River source region, we have more accurately estimated the $FCO_2$ by in situ measurement and discussed the

riverine $CO_2$ outgassing within four land cover types, but there are still many uncertainties in our research. Firstly, despite the slight increase in sampling sites compare with the previous studies, there was less extensive research on single watersheds that are spatially representative. And in terms of time, there was a lack of continuous sampling of long sequences. Existing research suggests that the rainstorms will have

a huge shift on $CO_2$ emission (Smits et al., 2017) and we lacked the monitoring of $CO_2$ outgassing during heavy rain period. These factors caused some uncertainties of the riverine $CO_2$ evasion research and high frequency and long-time sequence studies under specific land cover types need to be performed in the future.

## 5. Conclusions

Based on four rounds of field direct measurements of $CO_2$ outgassing within the Yellow River source
region, the average $pCO_2$ in the study area was estimated at $771\pm380$ µatm, and the average $FCO_2$ was $590\pm766$ g C m$^{-2}$ yr$^{-1}$. It is lower than other rivers in the world, and at a relatively low level compared to the middle and lower reaches of the Yellow River. The results showed that the rivers in the Yellow River source region were the source of $CO_2$. Both the $pCO_2$ and $FCO_2$ showed strong spatial and temporal variations. The largest $CO_2$ release from rivers was found in August, followed by October and April, and
the lowest was observed in June. When grouped into different land cover types. $FCO_2$ in the permafrost regions was the lowest among the four types of land cover. The highest $FCO_2$ Was found in peatland river, followed by grassland and glacier region.

In alpine climates, low temperature conditions played a crucial role in limiting biological activity and
reducing $CO_2$ emissions in the region. As a consequence, control both riverine $CO_2$ source and gas transfer velocity. The DOC has huge influence on all land cover types. In the permafrost region, the large amount soil related DOC could support riverine $CO_2$ concentration. In the glacier region, the glacial DOC and $CO_2$ may play an essential role in $CO_2$ outgassing. In the peatland and grassland region, the plants related DOC is an important source of riverine $CO_2$.


By integrating seasonal changes of water surface area, the $CO_2$ efflux was estimated at $0.37\pm0.49$ T g C



$yr^{-1}$, which is significantly different from the earlier estimate by Ran et al. (2015). Very few studies have focused on the dynamics of riverine carbon cycling on the Tibetan Plateau river systems. This study provides insight into the riverine $CO_2$ outgassing in the Yellow River source region, which will help

better understanding of carbon emissions from alpine rivers in the world, in particular these located on the Tibetan Plateau.

**Acknowledgements:** This work was supported by the Inner Mongolia University (grant: 30150-135114), the Natural Science Foundation of China (grants: 21800-5161101).

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



Table 1. Land cover types, altitude, stream types, pH, alkalinity, DOC, $pCO_2$, and $FCO_2$ of the 36 stream sites within the Yellow River source region, expressed in the order of April, June, August, and October.

| Name | Land cover types | Apr pH | Apr Alkalinity (mmol L$^{-1}$) | Apr DO (mg L$^{-1}$) | Apr DOC (mg L$^{-1}$) | Apr $pCO_2$ (µatm) | Apr $FCO_2$ (g C m$^{-2}$ yr$^{-1}$) | Jun pH | Jun Alkalinity (mmol L$^{-1}$) | Jun DO (mg L$^{-1}$) | Jun DOC (mg L$^{-1}$) | Jun $pCO_2$ (µatm) | Jun $FCO_2$ (g C m$^{-2}$ yr$^{-1}$) | Aug pH | Aug Alkalinity (mmol L$^{-1}$) | Aug DO (mg L$^{-1}$) | Aug DOC (mg L$^{-1}$) | Aug $pCO_2$ (µatm) | Aug $FCO_2$ (g C m$^{-2}$ yr$^{-1}$) | Oct pH | Oct Alkalinity (mmol L$^{-1}$) | Oct DO (mg L$^{-1}$) | Oct DOC (mg L$^{-1}$) | Oct $pCO_2$ (µatm) | Oct $FCO_2$ (g C m$^{-2}$ yr$^{-1}$) |
|---|---|---|---|---|---|---|---|---|---|---|---|---|---|---|---|---|---|---|---|---|---|---|---|---|---|
| M1 | grassland | 8.4 | 4.4 | 7.4 | 4.6 | 662 | 435 | 8.6 | 2.5 | 6.4 | 2.5 | 513 | 224 | 7.5 | - | 6.6 | 4.7 | 758 | 794 | 8.6 | 2.9 | 8.1 | - | 643 | 316 |
| M2 | glacier | 8.4 | 3.5 | 8.6 | 5.4 | 550 | 184 | 8.6 | 1.9 | 7.2 | 3.6 | 483 | 809 | 7.1 | - | 6.6 | 3.5 | 514 | 212 | 7.3 | 2.3 | 8.2 | 5.4 | 678 | 199 |
| M3 | grassland | 8.5 | 4 | 7.5 | 3.9 | 525 | 451 | 8.7 | 3.1 | 6.8 | 9.5 | 572 | 28 | 7 | - | 6.4 | 3.8 | 715 | 497 | 7.8 | 3.1 | - | 1.3 | 535 | 374 |
| GR1 | grassland | 7.9 | 1.7 | 8.6 | 3.7 | 886 | 307 | 8.2 | 2.1 | 6.6 | 1.2 | 538 | 184 | 7.6 | 2.4 | 7.1 | 3.6 | 1056 | 1603 | 7.3 | 1.8 | 7.4 | 1.4 | 830 | 589 |
| GR2 | grassland | 8.5 | 4.8 | 6.8 | 5.9 | 571 | 583 | 8.6 | - | 7.3 | 5.3 | 488 | 49 | 7.8 | - | 6.5 | - | 1095 | 1607 | 7.3 | 3.7 | 7.6 | 3.1 | 822 | 724 |
| GR3 | grassland | 8.2 | 2.8 | 7.1 | 10.0 | 686 | 297 | 8.2 | 2.1 | 8.6 | 5.1 | 1490 | 98 | 7.1 | 2.8 | 5.9 | 8.2 | 611 | 120 | 8 | 1.6 | 5.9 | 4.7 | 719 | 202 |
| GR4 | grassland | 8.6 | 4.2 | 6.7 | 3.6 | 614 | 175 | 8.7 | 3 | 6.7 | 5.4 | 479 | 71 | 7 | 2.7 | 6.0 | 6.9 | 552 | 524 | 7.3 | 3.9 | 8.0 | 6.6 | 510 | 1361 |
| GR5 | grassland | 8.4 | 4.4 | 7.5 | - | 969 | 1426 | 8.5 | 2.9 | 6.8 | 3.8 | 545 | 282 | 7.1 | 2.9 | 5.8 | 2.7 | 1197 | 1956 | 7.7 | 3.9 | 8.1 | 1.7 | - | - |
| GR6 | grassland | 8.4 | 5.6 | 8.4 | 4.7 | 762 | 500 | 8.5 | 3.7 | 7.6 | 9.6 | 628 | 727 | 7.2 | 3.9 | 6.3 | 8.4 | 1863 | 1619 | 7.3 | 5.1 | 8.4 | 4.2 | 1084 | 2174 |
| GR7 | grassland | 8.2 | 6.3 | 7.0 | 2.5 | 1393 | 889 | 8.2 | 3.2 | 6.5 | 3.0 | 608 | 438 | 7 | 3.5 | 6.9 | 2.8 | 2196 | 1478 | 7.2 | 5.5 | 8.2 | 1.7 | 1216 | 2321 |





| | | | | | | | | | | | | | | | | | | | | | | | | |
|---|---|---|---|---|---|---|---|---|---|---|---|---|---|---|---|---|---|---|---|---|---|---|---|---|
| GR8 | grassland | 8.2 | 2.6 | 8.3 | 7.4 | 720 | 199 | 8.8 | 1.9 | 9.0 | 7.8 | 478 | 331 | 7.2 | 2.8 | 7.3 | 2.6 | 1416 | 923 | 7.2 | 2.8 | 7.4 | 5.3 | 593 | 457 |
| GR9 | grassland | 8.1 | 2.3 | 8.7 | 4.8 | 921 | 138 | 8.5 | 1.6 | 7.0 | 6.8 | 493 | 641 | 7.1 | 1.7 | 6.4 | 2.0 | 585 | 524 | 7.2 | 1.9 | 7.6 | 6.7 | 515 | 533 |
| GR10 | grassland | 8 | 3.4 | 8.0 | 3.5 | 1124 | 1036 | 8.6 | 2.2 | 6.7 | 2.3 | 469 | 454 | 7.1 | 1.9 | 6.2 | 2.7 | 676 | 175 | 7.3 | 2.4 | 7.9 | 2.4 | 496 | 236 |
| GR11 | grassland | 8.5 | 4.6 | 7.9 | 4.9 | 547 | 230 | 8.7 | 3 | 6.9 | - | 482 | 178 | 7.2 | - | 6.2 | 6.6 | 700 | 402 | 7.8 | 4.1 | 7.7 | - | 557 | 441 |
| Pt1 | peatland | 8.6 | 4.8 | 7.3 | 6.3 | 562 | 448 | 8.6 | 3.4 | 6.0 | 4.5 | 522 | 107 | 7 | 3.6 | 6.5 | 2.5 | 1970 | 1358 | - | 3.5 | 7.1 | 5 | 876 | 1193 |
| Pt2 | peatland | 8.2 | 5 | 8.5 | 6.3 | 1362 | 1128 | 8.2 | 3.3 | 8.3 | 3.0 | 598 | 61 | 7.1 | 1.8 | 6.5 | 6.1 | 1461 | 328 | 7.3 | 2.6 | 7.6 | 3.8 | 862 | 1251 |
| Pt3 | peatland | 7.4 | 1 | 7.6 | - | 1809 | 831 | 7.9 | - | 6.6 | 1.4 | 1139 | 527 | 7.4 | 1.1 | 6.4 | 2.4 | 862 | 6892 | 8 | 1.2 | 7.4 | 6.8 | 1370 | 1818 |
| Pt4 | peatland | 8.4 | 5.9 | 8.4 | 5.3 | 511 | 383 | 8.6 | 3.5 | 8.2 | 2.6 | 509 | 251 | 7.3 | 3.5 | 7.5 | 3.8 | 882 | 129 | 7.3 | 4.2 | 8.8 | 0.2 | 517 | 175 |
| Pt5 | peatland | 7.7 | 1 | 7.0 | 5.1 | 576 | 282 | 8.6 | 1.5 | 7.1 | 7.6 | 660 | 273 | 7.1 | 1.8 | 6.3 | 4.2 | 523 | 92 | 7.2 | 1.7 | 7.8 | 4.4 | 856 | 650 |
| Pt6 | peatland | 7.5 | 0.6 | 8.0 | 4.3 | 1177 | 1015 | 7.9 | 1.2 | 6.3 | 9.4 | 652 | 221 | 7.1 | 0.8 | 5.9 | 3.1 | 632 | 546 | 7.6 | 0.9 | 7.0 | 4.6 | 1206 | 1515 |
| Pt7 | peatland | 7.6 | 0.7 | 8.2 | - | 612 | 1469 | 8.2 | 1.5 | 7.0 | - | 490 | 267 | 7.2 | 1.3 | 5.5 | 3.5 | 755 | 1778 | 7.5 | 1 | 8.0 | 3.2 | 685 | 123 |
| Pt8 | peatland | 7.6 | 1.1 | 7.8 | 6.3 | 712 | 454 | 8.2 | 1.3 | 6.4 | - | 567 | 172 | 7.2 | 1.2 | 6.2 | 2.1 | 2441 | 1226 | 7.9 | 1.2 | 7.6 | 3.9 | 732 | 2030 |
| Pt9 | peatland | 8.6 | 4.8 | 8.1 | 6.6 | 562 | 34 | 8.1 | - | - | 12.2 | 891 | 307 | 7.1 | 2.1 | 7.5 | 21.7 | 1268 | 61 | 7.3 | - | 7.1 | 5.1 | 1338 | 346 |
| Pt10 | peatland | 7.8 | 1 | 7.1 | 5.1 | 865 | 126 | 8.5 | 1.6 | - | 3.0 | 1891 | 89 | 7.1 | - | 7.5 | 3.4 | 761 | 233 | - | 1.1 | 7.2 | - | 815 | 478 |
| GL1 | glacier | 8.4 | 3.9 | 7.2 | 4.4 | 656 | 1190 | - | - | - | - | - | - | 7.6 | 1.2 | 8.3 | 2.1 | 273 | -144 | 7.2 | 2.9 | 9.8 | 4.8 | 806 | 454 |
| GL2 | glacier | 8.2 | 1.8 | 8.7 | 4.7 | 859 | 1318 | 8.6 | - | 7.5 | 2.0 | 484 | 83 | 7.5 | - | 6.6 | - | 628 | 392 | 7.5 | - | 7.7 | - | 711 | 399 |
| GL3 | glacier | 8.5 | 3.1 | 8.2 | 4.7 | 606 | 156 | 8.7 | 2.3 | 6.8 | 2.8 | 492 | 52 | 7.3 | 1.9 | 6.4 | 4.5 | 692 | 583 | 7.8 | 2.5 | 8.0 | 3.9 | 523 | 230 |
| GL4 | glacier | 8.4 | 3.1 | 8.1 | 2.7 | 514 | 205 | - | - | - | - | - | - | 7.5 | 2.1 | 6.6 | 4.5 | 716 | 328 | 7.8 | 3.1 | 8.4 | 4.6 | 672 | 328 |





| | | | | | | | | | | | | | | | | | | | | | | | | | |
|---|---|---|---|---|---|---|---|---|---|---|---|---|---|---|---|---|---|---|---|---|---|---|---|---|---|
| GL5 | glacier | 8.5 | 3.1 | 7.5 | 4.1 | 630 | 46 | - | - | - | - | - | - | 7.3 | 2.2 | 6.9 | 2.9 | 525 | 1119 | 7.3 | - | 7.8 | - | 441 | 175 |
| GL6 | glacier | 8.5 | 4 | 7.2 | 3.9 | 542 | 405 | 8.6 | - | 6.9 | 1.2 | 542 | 264 | 7.3 | 2.8 | - | 2.2 | 1592 | 2459 | - | 2.3 | 7.9 | 4 | 640 | 632 |
| Pm1 | permafrost | 8.8 | 3.2 | - | 3.4 | 236 | -141 | 10.6 | 7.4 | 12.1 | - | - | - | 7.3 | 2.9 | 6.7 | 6.5 | 560 | 18 | 7.2 | - | 8.5 | - | 936 | 101 |
| Pm2 | permafrost | 8.6 | 4.1 | - | 5.5 | 511 | 426 | 8.7 | - | 6.2 | 5.1 | 483 | 138 | 7.2 | 3 | 5.7 | 8.2 | 538 | 270 | 8.1 | 7.6 | 6.4 | - | 927 | 933 |
| Pm3 | permafrost | 8.8 | 5.2 | - | 6.3 | 681 | 267 | 9 | - | 7.2 | 5.0 | 447 | 44 | 7.2 | - | 6.9 | 7.1 | 554 | 149 | - | 4.1 | 7.2 | 3.8 | 365 | -34 |
| Pm4 | permafrost | 8.3 | 3.2 | 8.0 | 2.8 | 495 | 254 | 8.5 | - | 6.7 | 1.5 | 508 | 126 | 7.2 | - | 6.4 | 2.5 | 628 | 316 | - | 2.2 | 8.4 | 1.1 | 583 | 227 |
| Pm5 | permafrost | 8.3 | 4.3 | 7.7 | 4.6 | 688 | 322 | 8.7 | 3.4 | 6.9 | 4.8 | 502 | 374 | 7.2 | 3.1 | 7.4 | 7.1 | 726 | 757 | 7.2 | 4.4 | 8.0 | 10.6 | 859 | 806 |
| Pm6 | permafrost | 8.3 | 2.1 | 7.3 | 7.4 | 181 | -221 | 8.5 | 2.5 | 6.2 | 4.6 | 989 | 478 | 7.3 | 1.8 | 9.2 | 6.5 | 540 | 98 | - | 2.6 | 7.0 | 1.4 | 866 | 1233 |





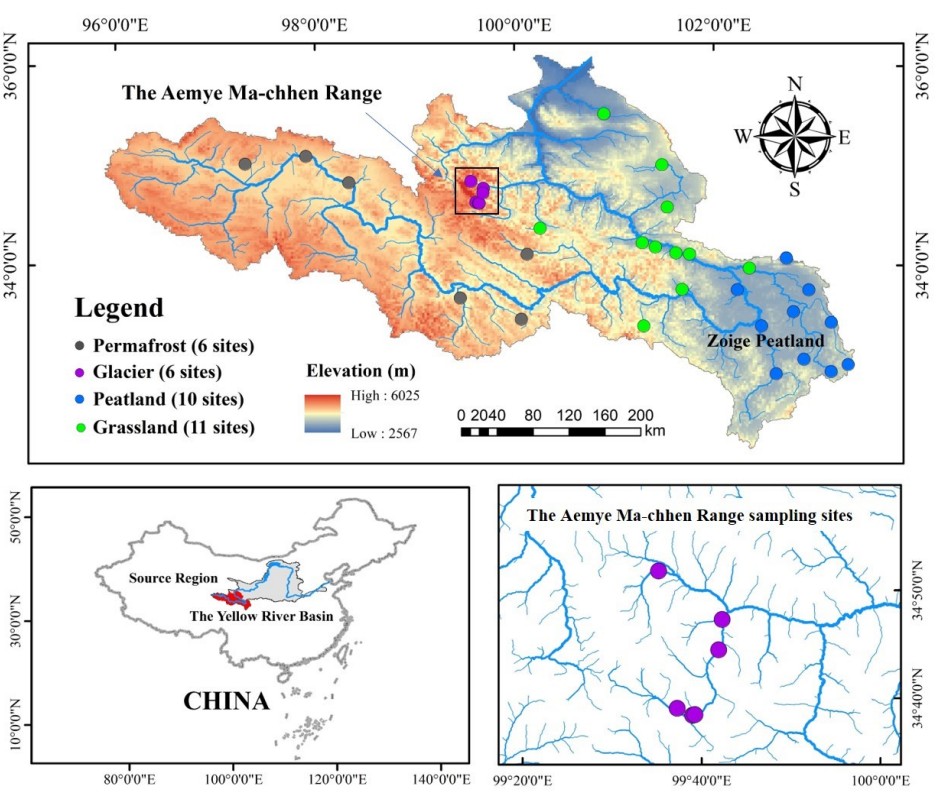

Figure 1. Sampling sites of the Yellow River Source Region





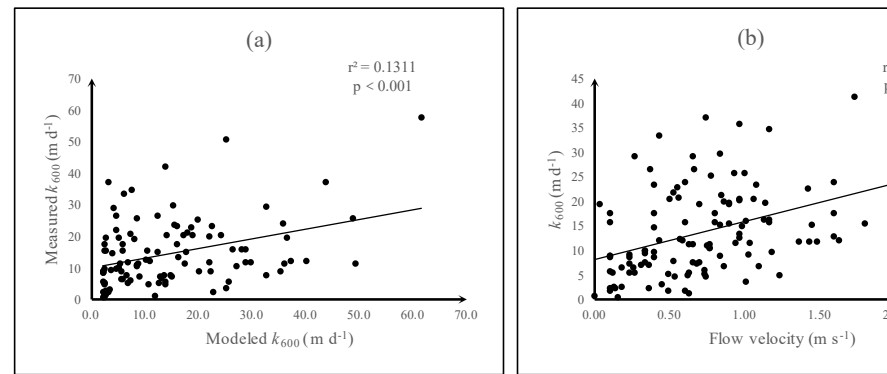

Figure 2. (a) The relationship between actual and predicted $k_{600}$ for streams; (b) Correlation between standardized gas transfer velocity ($k_{600}$) and flow velocity over the 4 campaigns. High $k_{600}$ values (>70 m d$^{-1}$) were removed from analysis.



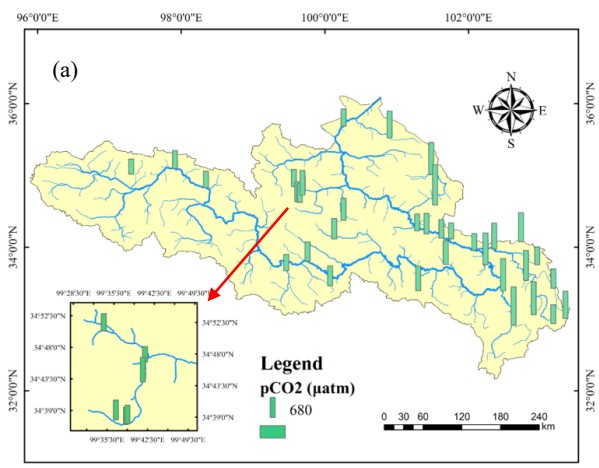

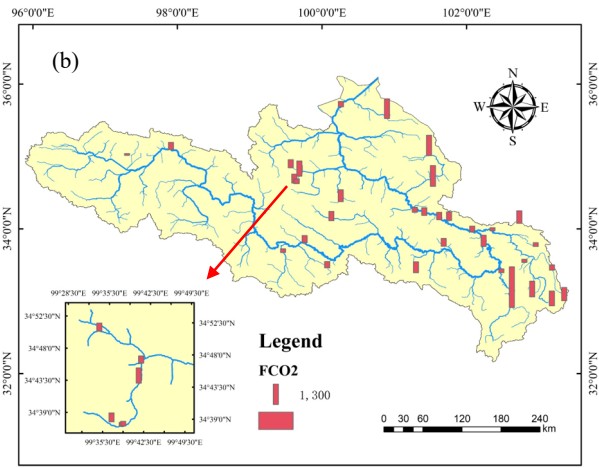

Figure 3. Spatial and temporal variations of average $p\text{CO}_2$ (3a) and $F\text{CO}_2$ (3b) within the Yellow River source region.





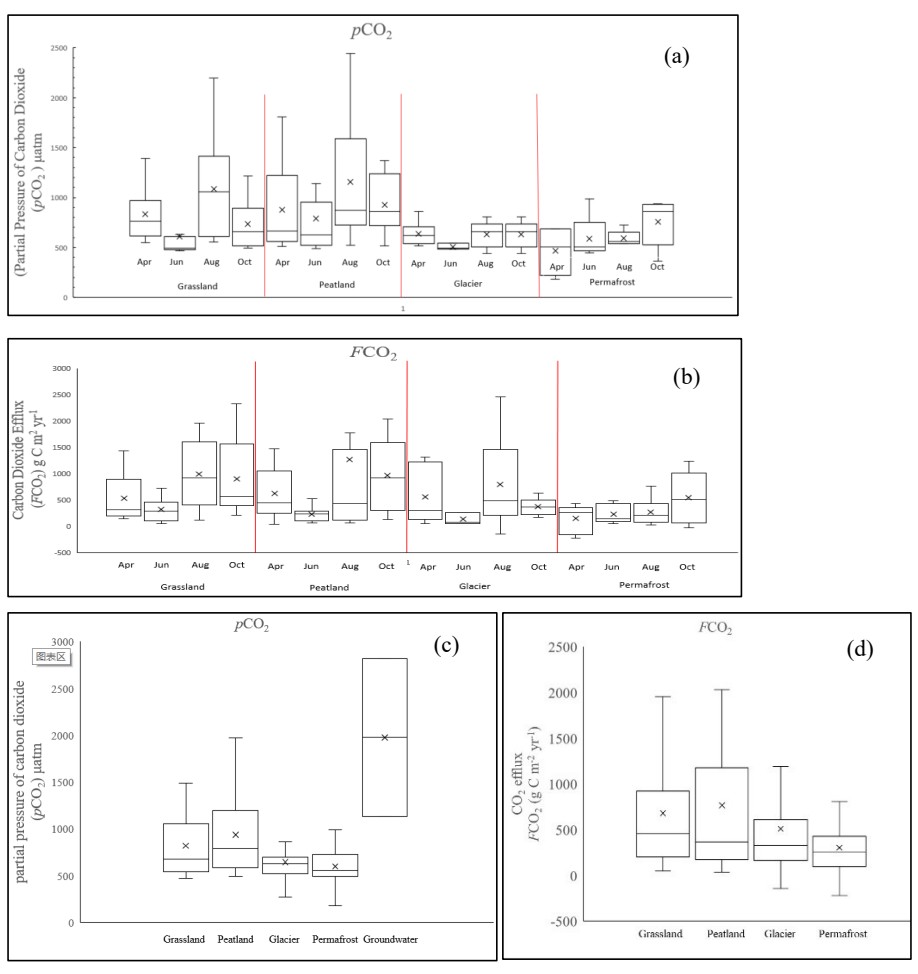

Figure 4. The box plots of $p$CO$_2$ and $F$CO$_2$ under four different land cover types within the Yellow River source region,

expressed in the order of April, June, August, and October in Figure 4a, 4b. The $p$CO$_2$ data expressed in the order of

grassland, peatland, glacier, permafrost, and groundwater in figure 4c, The $F$CO$_2$ expressed in the order of grassland,

peatland, glacier, and groundwater in figure 4d.




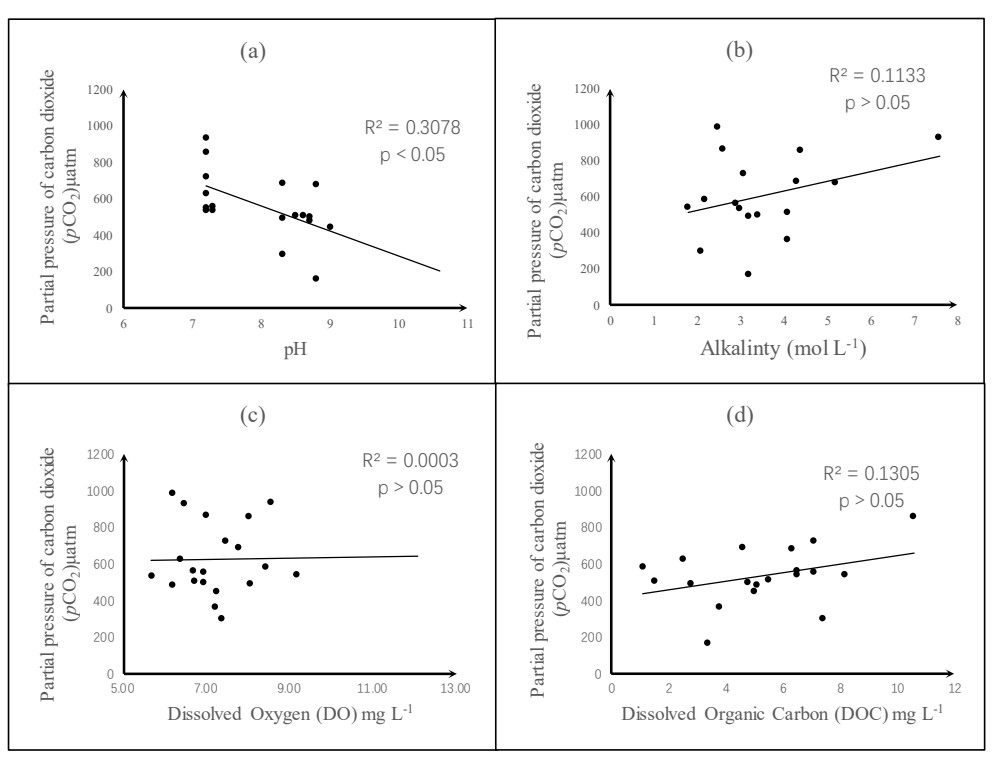

Figure 5. The linear relationship of chemical parameters and $p\mathrm{CO_2}$ in permafrost covered region. (a) pH, (b) alkalinity,

(c) dissolved oxygen, and (d) dissolved organic carbon.





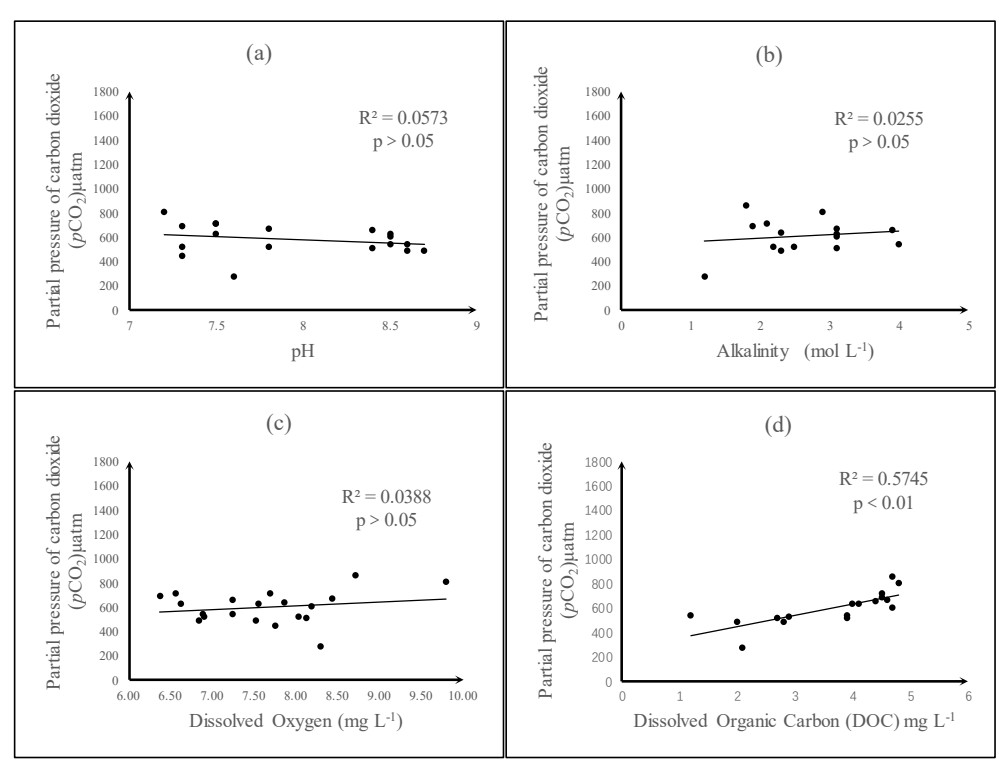

Figure 6. The linear relationship of chemical parameters and $p\mathrm{CO_2}$ in glacier covered region. (a) pH, (b) alkalinity, (c)

dissolved oxygen, and (d) dissolved organic carbon.



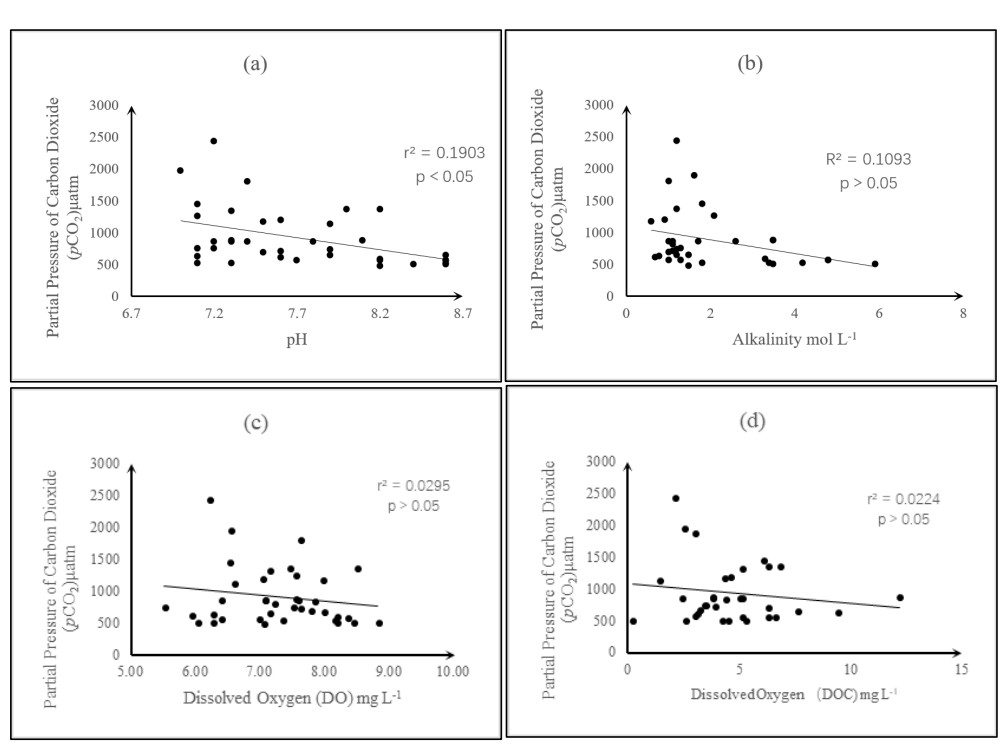

Figure 7. The linear relationship of chemical parameters and $p\mathrm{CO_2}$ in peatland covered region. (a) pH, (b) alkalinity, (c) dissolved oxygen, and (d) dissolved organic carbon.





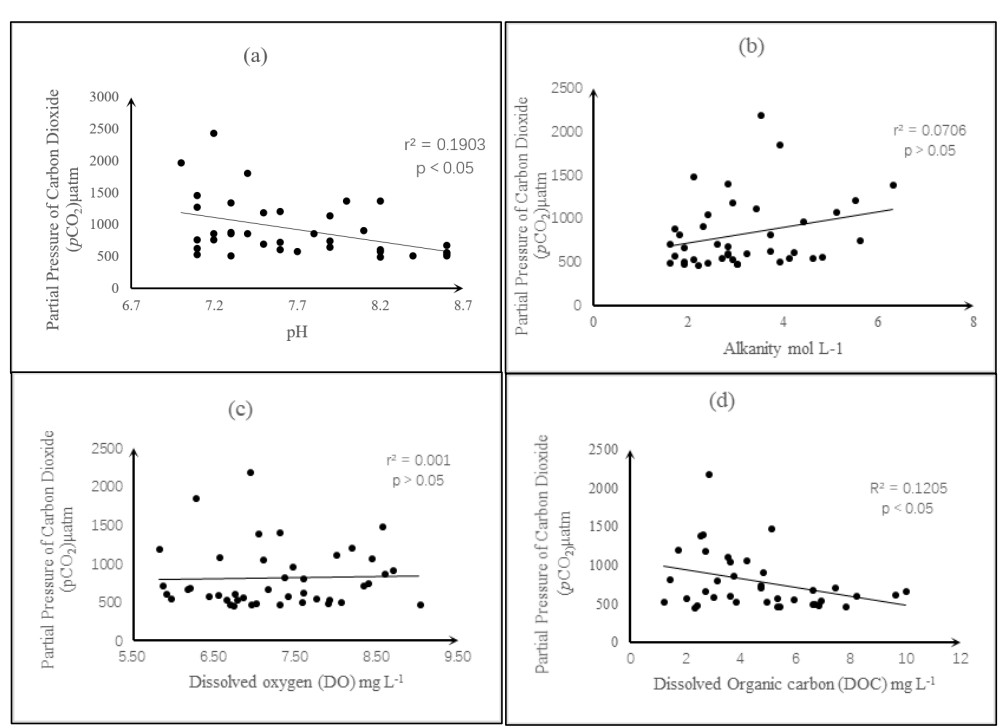

Figure 8. The linear relationship of chemical parameters and $p$CO$_2$ in grassland covered region. (a) pH, (b) alkalinity, (c) dissolved oxygen, and (d) dissolved organic carbon.