# Peer review of "High Riverine CO2 Outgassing affected by Land Cover Types in the Yellow River Source Region"

_Biogeosciences, 2018_

## Referee Comment (RC1) · Anonymous Referee #1 · 14 Sep 2018

Major comments

The procedure to compute pCO2 with the equation in L202 from the headspace measurements (prior and after equilibration) is incorrect and does not correspond to the one described by Dickson et al. (2007) (as stated by the authors). The major problem in the approach proposed in this equation is that it does not take into account the buffering capacity (due to the presence of HCO3-, or alkalinity) in the water sample. So for a same pCO2 in water (true value) and a same pCO2 in air (initial value prior to shaking), the final pCO2 in the headspace will be very different depending on the alkalinity of the sample. If we imagine a theoretical case of a near-infinite alkalinity

water sample, then the final pCO2 in the headspace will be nearly quasi-identical to the pCO2 in water: due to the near-infinite buffering capacity, the solution will be able to adjust for the equilibration of the headspace and the water sample. This will not be the case of a zero alkalinity sample, for which the final pCO2 in the headspace will be intermediate between the "true" value and the pCO2 in air prior to shaking.

Dickson et al. (2007) give in SOP4 (Determination of pCO2 in air that is in equilibrium with a discrete sample of sea water) a procedure that relies on the readjustment of DIC to take into account the change of pCO2 in the headspace that allows to re-compute "initial" pCO2 from DIC and alkalinity, once DIC is corrected. Since the authors have alkalinity data, they have all of the data to make these computations that are easily achievable with a software to compute the CO2 speciation such as CO2SYS.

Also, the procedure to compute pCO2 should take into account temperature variations between in-situ temperature and the final temperature at which equilibrium was achieved. It's unclear how this was done and at which temperature the K0 in the equation L202 was computed.

As the paper stands, I do not fully trust the pCO2 data presented due to unclear computation procedure.

Indeed, a systematic over-estimation of pCO2 values could explain a systematic under-estimation in the computation of k600 values that could explain why the computed k600 values are lower than those modelled (based on a parameterisation derived from tracer experiments) (Fig. 2). Indeed, the exact contrary would be expected since floating chamber measurements with a fixed chamber lead to an enormous over-estimation of the flux measurements (Lorke et al. 2015).

Specific comments

L 20 : I'm not sure that the large uncertainty on the estimate of riverine CO2 emission is due to a lack of data "especially" in headwaters (as stated). There is a generalised lack

of high quality CO2 data everywhere in rivers. Given that 80% of riverine emissions of CO2 are in the tropics, I would assume that largest source of uncertainty is lack of data in tropical areas.

L 45: Please rephrase. Researchers do not "believe". They build theories and test hypotheses.

L46: Emerging evidence (Abril et al. 2014) points to the importance of wetlands in driving riverine CO2 emissions rather than "terrestrial ecosystems", as these are conceptually difference sources of carbon for rivers.

L 55: Lauerwald et al. (2015) gives a global estimate of 0.65 PgC yr-1.

L55: The value of 0.7 PgC/yr from Cole et al. (2007) is for all inland waters. Cole et al. (2007) give a value of 0.2 PgC/yr for rivers alone.

L 64: studies instead of studied

L78-79: can you please develop the differences of "underlying control mechanisms" for CO2 emissions between alpine climate and other climates? For me it's the same mechanisms, but it's just colder.

L93-1002: Ran et al. (2017) reports very extensive data-set in the Wuding River (part of the Yellow river basin) with data obtained at altitudes up to 1340m. So, some information is already available.

L 155: The determination of end-point with Methylorange indicator seems a bit crude. Can you please state the estimated accuracy of the method ? Did you check the accuracy with standards made of NaCO3 ?

L 245: You cannot conclude that the dampening effect of chambers is responsible for the lack of correlation between wind and K600. If k600 is overwhelmingly driven by other processes than wind, then you would also arrive at a lack of correlation irrespective of a dampening effect.

L 252: clarify statement "mainly due to the water temperature played a crucial role in limiting CO2 transfer between the air-water interface". Since the gas transfer is normalized to Schmidt number of 600, this automatically removes the effect of water temperature.

L 303: the relationship between pCO2 and DOC can also indicate that both have a common origin such as (simultaneous) inputs from soils. It does not imply that DOC "supports" CO2 production as stated.

L 349: Abril et al. (2014) report on the influence of floodplain lakes on CO2 dynamics in the Amazon rivers, not "peatland rivers" as stated.

L 349: Hu et al. 2015 is missing from the reference list. I assume it's Hu 2005, that seems to correspond to a PhD thesis from one of the co-authors, possibly based on part of the data reported in this paper, so corresponding to circular auto-citation.

L375: rephrase "Although groundwater is participated"

Legend of figure 2: On what grounds did you exclude k600 data above 70 m/d ? It is very awkward to exclude data without justification.

In figures 5-8 it could be useful to plot pCO2 versus water temperature.

References

Abril G., Martinez J.-M., Artigas L.F., Moreira-Turcq P., Benedetti M.F., Vidal L., Meziane T., Kim J.-H., Bernardes M.C., Savoye N., Deborde J., Albéric P., Souza M.F.L., Souza E.L. and Roland F. (2014) Amazon River carbon dioxide outgassing fuelled by wetlands. Nature 505: 395–398.

Lauerwald, R. et al. Spatial patterns in CO2 evasion from the global river network, Global Biogeochem. Cycles, 29, 534–554, doi:10.1002/2014GB004941. 2015.

Lorke, A., Bodmer, P., Noss, C., Alshboul, Z., Koschorreck, M., Somlai-Haase, C., Bastviken, D., Flury, S., McGinnis, D. F., Maeck, A., Müller, D., and Premke, K.: Technical note: drifting versus anchored flux chambers for measuring greenhouse gas emissions from running waters, Bioge osciences, 12, 7013-7024, 2015.

---

## Referee Comment (RC2) · Anonymous Referee #2 · 22 Sep 2018

The manuscript by Mingyang Tian and co-authors investigates the riverine partial pressure of CO2 and CO2 efflux in the source region of the Yellow River and differenciates between landscape types (glacier, permafrost, wetland and grassland). This approach is different than the most studies about CO2 in rivers. Commonly studies compare streams/rivers by size or by climate zone (Marx et al. 2017, Lauerwald et al. 2015, Raymond et al. 2013). Thus, this study aims to improve the understanding of carbon dioxide emissions in alpine rivers, particularly in the Tibetan Plateau. Further, different methods to determine CO2 degassing were applied: floating chambers and headspace equilibrium method. Unfortunately, the uncertainties of results from the different methods are hardly discussed.

Overall, in my opinion the work has a high potential to make a good contribution to the understanding of CO2 emission in rivers. However, at present several aspects require extensive revision. That is why I cannot recomment this manuscript for publication in its present form. I recommend re-submission after thourough revision according to the points below.

Major comments:

(1) This promising paper is restricted to the Yellow River source regions and lacks moving beyond that. I recommend showing results in a global context.

(2) The final version of the paper would benefit from editing for language. While it is generally understandable, several idiomatic expressions and mistakes hamper the readability. For instance, make sure the text is in past-tense and use "the" before plural and delete it if singular follows.

(3) I strongly recommend to discuss the reliability of your data. For instance, how were fluxes determined by chamber method compared to fluxes by equilibrium headspace method?

Specific comments:

Abstract

Ln. 1: Don't use the same sentence to start the abstract and the introduction

Ln. 25: They mostly were a CO2 source. I remember that you also showed negative fluxes.

Ln. 32: Be careful with "significant" as it indicates a statistical significance. Is that the case here? Otherwise replace.

Ln. 32-33: Rephrase sentence it is difficult to understand.

Ln. 33-37: These sentences include general knowledge. Give some numbers or statements derived from you research, as this increases the impact of the paper.

Introduction

Ln. 45: Researcher should not believe. General agreement/ consensus/ ...?

Ln. 48-52: Please revise sentence as it is hardly readable.

Ln. 55-57: Check Swakuchi et al. 2017. They included lower reaches of the amazon to global estimate by Raymond et al. (2013). This led to values of 2.58 petagrams (Pg) CO2 yr-1 for rives and streams. See also Marx et al. (2017).

Ln 57-60: This sentence is not clear. Do you mean there is a lack of direct measurement data? That is true. The studies you mention calculate pCO2 from DIC/Alkalinity, pH and T, as this is the common method. There is a decent database (Hartmann et al. 2014), that was basis for global CO2 emission estimates (Raymond et al. 2013; Lauerwald et al. 2015).

Ln. 61: What do you mean by global river systems?

Ln. 73: Wrong word. This was not "concluded" but determined/measured.

Ln. 86: Statistically significant? Can you give a number?

Ln. 87-88: Do you compare autumn values here?

Ln. 102: In the alpine rivers of the Yellow River?

Ln. 107-112: Good!

Material and methods

Later in the text you mention carbonate rocks/limestone. I recomment to provide data on bedrock geology in this section.

Ln. 138: analyses?

Ln. Accuracy of 0.004 for pH measurements in the field. Is that really realistic?

Ln. 150: This is basic knowledge: inorganic carbon species distribution dependency on pH (and temperature).

Ln. 153: Be consistent and add the country behind the company.

Ln. 57: "Specific bottle" be more precise! Brown glass bottle? Volume?

Ln 159: Germany

Ln. 160: Not less. "... precision better than ..."

Ln. 162: Use Determination instead of calculation here.

Ln. 164-165: Can you give a precision of the Li-7000 analyzer?

Ln. 175-176: Were 6-10 mins sufficient to get data for a linear flux estimation?

Ln. 181-190: A better describtion is needed here. The first equation is a linear approximation of CO2 flux from chamber measurements. The second is mostly applied when pCO2 is calculated from DIC/Alkalinity, pH and T. However, here it can be used to estimate kH.

Ln. 200: "... blow $\pm$ 3

Ln. 195-208: Good method. Did you ever apply chambers until equilibrium was reached? It would be interesting to see if results match.

Results

Ln. 218-219: Please add the year.

Ln. 229: The last part of the sentence is interpretation and belongs to the discussion section.

Ln. 244: This is new. You did not mention with in the MaterialMethods section! I thought you determined k600 by rearranging equation (2)? This belongs to the MaterialMethods section.

Ln. 245: Computed? From what? Chamber measurements? I guess with model results you refer to model 5 by Raymond et al. (2012).

Ln. 246: This belongs to the discussion, not results.

Ln. 249-251: This sentence is not understandable. Please rephrase. Be more precise: did you apply a statistical approach to determine relationships?

Ln. 251-255: This belongs to the discussion, not results.

Ln. 279: Statistically significant? Otherwise don't use "significant".

Discussion

Ln. 294: This is a poor beginning. Better describe the key result in a larger context "This study shows/demonstrates..." to create a red line for the forthcoming discussion.

Ln. 295: Be aware that not all the riverine CO2 is derived from land. Your statement is not correct.

Ln. 310-312: Please revise this sentence.

Ln.314: Replace "not easy".

Ln. 325-326: This sentence is vague and insignificant. Rephrase with details.

Ln. 326: Use past-tense.

Ln. 327-229: Rephrase sentence to increase readability.

Ln. 335: "highlights"

Ln. 341-344: Rephrase this sentence.

Ln. 346-...: Use past-tense.

Ln. 355: Sufficient for what?

Ln. 366: "Analyses"

Ln. 367-368: The relationship between pCO2 and pH is well known. This sentence is not correct, as the pH (and T) determines the species of inorganic carbon in water.

Ln. 375-381: What makes you think that groundwater input is higher in grassland regions? If you give a statement like this, you need references. Groundwater samples are not sufficient evidence for this statement, as groundwater pCO2 typically are higher than stream pCO2.

Ln. 384: see comment Ln. 294:

Ln. 401-403: There are several other potential reasons. How about pH changes (higher pH means less carbon in the form of CO2)(Stets et al. 2017)? And how about decreasing proportion of groundwater distribution with increasing stream/river size (Marx et al. 2017)?

Ln. 411: "Easily neglected"?

Conclusions

Ln. 445: Revise sentence. Verb missing?

Ln. 451: Is this flux for the study area or the whole Yellow River?

Ln. 452: What is the number in Ran et al.? Is the number for the whole Yellow River?

Table

Add the year somewhere. The table has to be understandable for itself. Revise subscripts and superscripts, as there are many mistakes.

Figure

Figure 1: Good!

Figure 2: (a) Are these mean values for all your data? Add a small paragraph in the MaterialMethods section where you explain your values? Use same names in the Figure title than for the axes labels. Add reference for modeled k600 (Raymond?). (b)

Which k600 did you display here? Measured or modeled ones? Please clearify. The last sentect in your Figure title should be explained in the MaterialMethods section.

Figure 3: Add the dimension for fluxes. What for is the box (red, green) under the legend? Delete? Figure 4: Write "Figure" with a capital letter at the beginning. Add the year in the Figure title.

Figure 5 and 6: Be consistent with brackets: Dimensions either inside or outside brackets.

Figure 7: (d) the a-axis label is not correct.

References

Hartmann, J., R. Lauerwald, and N. Moosdorf (2014), A brief overview of the Global River Chemistry database, GLORICH, Proc. Earth Planet. Sci.,10, 23–27.

Marx, A., J. Dusek, J. Jankovec, M. Sanda, T. Vogel, R. van Geldern, J. Hartmann, and J. A. C. Barth (2017), A review of CO2 and associated carbon dynamics in headwater streams: A global perspective, Rev. Geophys., 55, doi:10.1002/2016RG000547.

Lauerwald, R., G. G. Laruelle, J. Hartmann, P. Ciais, and P. A. G. Regnier (2015), Spatial patterns in CO2 evasion from the global river network, Global Biogeochem. Cycles, 29, 534–554, doi:10.1002/2014GB004941.

Raymond, P. A., et al. (2013), Global carbon dioxide emissions from inland waters, Nature, 503(7476), 355–359.

Sawakuchi, H. O., et al. (2017), Carbon dioxide emissions along the lower Amazon River, Front. Mar. Sci., 4(76).

Stets, E. G., D. Butman, C. P. McDonald, S. M. Stackpoole, M. D. DeGrandpre, and R. G.

Striegl (2017), Carbonate buffering and metabolic controls on carbon dioxide in rivers,

Global Biogeochem. Cycles, 31, 663–677.

---

## Author Comment (AC1) · 22 Oct 2018

Dear reviewer,

We are grateful to your comments on the manuscript. Based on your very constructive comments, we have thoroughly revised the manuscript. We have also responded below to all your comments. Please see below the details. Major revisions have also been highlighted in the revised manuscript in green color.

With best regards
Mingyang Tian, on behalf of the coauthors

**Major comments**

The procedure to compute $pCO_2$ with the equation in L202 from the headspace measurements (prior and after equilibration) is incorrect and does not correspond to the one described by Dickson et al. (2007) (as stated by the authors). The major problem in the approach proposed in this equation is that it does not take into account the buffering capacity (due to the presence of $HCO_3^-$, or alkalinity) in the water sample. So, for a same $pCO_2$ in water (true value) and a same $pCO_2$ in air (initial value prior to shaking), the final $pCO_2$ in the headspace will be very different depending on the alkalinity of the sample. If we imagine a theoretical case of a near-infinite alkalinity water sample, then the final $pCO_2$ in the headspace will be nearly quasi-identical to the $pCO_2$ in water: due to the near-infinite buffering capacity, the solution will be able to adjust for the equilibration of the headspace and the water sample. This will not be the case of a zero-alkalinity sample, for which the final $pCO_2$ in the headspace will be intermediate between the "true" value and the $pCO_2$ in air prior to shaking.

Dickson et al. (2007) give in SOP4 (Determination of $pCO_2$ in air that is in equilibrium with a discrete sample of sea water) a procedure that relies on the readjustment of DIC to take into account the change of $pCO_2$ in the headspace that allows to re-compute "initial" $pCO_2$ from DIC and alkalinity, once DIC is corrected. Since the authors have alkalinity data, they have all of the data to make these computations that are easily achievable with a software to compute the $CO_2$ speciation such as CO2SYS. Also, the procedure to compute $pCO_2$ should take into account temperature variations between in-situ temperature and the final temperature at which equilibrium was achieved. It's unclear how this was done and at which temperature the K0 in the equation L202 was computed. As the paper stands, I do not fully trust the $pCO_2$ data presented due to unclear computation procedure. Indeed, a systematic over-estimation of $pCO_2$ values could explain a systematic underestimation in the computation of k600 values that could explain why the computed k600 values are lower than those modelled (based on a parameterisation derived from tracer experiments) (Fig. 2).

Reply: We apologize for using this ambiguous equation as stated to mislead readers. We have provided a more detailed description of the equations (Lines 200 to 209). In fact, we have already considered the influence of alkalinity and have cited the method developed by Dickson et al. (2007). We have also considered the salinity indicator of fresh-water river systems and the calibration procedures of water vapor pressure. The reactions that take place

when carbon dioxide dissolves in water can be represented by the following equilibria:

$$CO_2(g) \rightleftarrows CO_2(aq) \tag{1}$$

$$CO_2(aq) + H_2O \rightleftarrows H_2CO_3(aq) \tag{2}$$

$$H_2CO_3(aq) \rightleftarrows H^+(aq) + HCO_3^-(aq) \tag{3}$$

$$HCO_3^-(aq) \rightleftarrows H^+(aq) + CO_3^{2-}(aq) \tag{4}$$

The notations (g), (l), and (aq) refer to the state of the species, *i.e.*, a gas, a liquid, and in aqueous solution, respectively. The sum of the $CO_2$ (aq) and $H_2CO_3$(aq) concentrations is expressed as $CO_{2*}$ (aq).

Redefining (1), (2), and (3) in terms of this species gives (5) and (6)

$$CO_2(g) \rightleftarrows CO_2^*(aq) \tag{5}$$

$$CO_2^*(aq) + H_2O(l) \rightleftarrows H^+(aq) + HCO_3^-(aq) \tag{6}$$

The equilibrium relationships between the concentrations of these various species can then be written as

$$K_0 = [CO_2^*]/pCO_2 \tag{7}$$

$$K_1 = [H^+][HCO_3^{2-}]/[CO_2^*] \tag{8}$$

$$K_2 = [H^+][CO_3^{2-}]/[HCO_3^-] \tag{9}$$

The calculation of $\ln(K/k^\circ)$ is given by the expression (10) below (Weiss, 1974):

$$\ln(K/k^\circ) = 93.4517 \left(\frac{100}{T/K}\right) - 60.2409 + 23.3585 \ln\left(\frac{T/K}{100}\right)$$

$$+S[0.023517 - 0.023656(\tfrac{T/K}{100}) + 0.0047036 \left(\tfrac{T/K}{100}\right)^2] \tag{10}$$

The calculation of $\log_{10}(K_1/k^\circ)$ is given by the expression below (Lueker et al., 2000):

$$\log_{10}(K_1/k^\circ) = \frac{-3633.86}{(T/K)}) + 61.2172 - 9.67770 \ln(T/K)$$

$$+0.011555S - 0.0001152S^2 \tag{11}$$

The calculation of $\log_{10}(K_2/k^\circ)$ is given by the expression below (Lueker et al., 2000):

$$\log_{10}(K_2/k^\circ) = \frac{-471.78}{(T/K)}) - 25.9290 + 3.16967 \ln(T/K)$$

$$+0.01781S - 0.0001122S^2 \tag{12}$$

Where, $k^\circ$ =1 mol kg-soln$^{-1}$, T is the temperature of the water, K is the kelvin of the water, S is the salinity.

The dissolved inorganic carbon content of water is defined as (13)

$$C_T = [CO_2^*] + [HCO_3^-] + [CO_3^{2-}] \tag{13}$$

Redefining (7), (8), and (9) in terms of this species gives (13)

$$= pCO_2^{headspace,f} \times K_0 \times [1 + \frac{K_1}{[H^+]} + \frac{K_1 \cdot K_2}{[H^+]^2}] \tag{14}$$

Where, the brackets represent total concentrations of these constituents in solution (in mol kg$^{-1}$) and [CO$_2$*] represents the total concentration of all unionized carbon dioxide, whether present as $H_2CO_3$ or as $CO_2$.

The $CO_2$ in water that emits into the headspace during the shaking process can be express:

$$D_C = \frac{Vh}{Vw}(pCO_2^{headspace,f} - pCO_2^{headspace,i})/(RT) \tag{15}$$

Redefining (13), the original $pCO_2$ of water could be calculated by (16)

$$pCO_2 = \frac{C_T + D_C}{[1 + \frac{K_1}{[H^+]} + \frac{K_1 \cdot K_2}{[H^+]^2}] \cdot K_0} \tag{16}$$

Finally, the $pCO_2$ was corrected by water vapor pressure

$$pCO_2^{Correct} = pCO_2^{Dry}(1 - pH_2O) \tag{17}$$

Where, $pCO_2^{Dry}$ is the corrected $pCO_2$ value in dry air, $pH_2O$ is the water vapor pressure over a water sample of given salinity at the temperature of equilibration, $pCO_2^{Correct}$ is the final corrected $pCO_2$.

Below is a screenshot of our calculation process shown as Figure 1.

[Figure]

Figure 1. The screenshot of raw $pCO_2$ data calibration

Indeed, the exact contrary would be expected since floating chamber measurements with a fixed chamber lead to an enormous over-estimation of the flux measurements (Lorke et al. 2015).

Reply: Yes, there is an enormous over-estimation of the static chamber method. In fact, for large rivers with relatively favorable flow conditions (>2.5 m wide), we tied the chamber to a small rubber boat and freely drifted along the river course to measure the $F$CO$_2$. Over the 36 sites, 32 (90%) of which we deployed the freely drifting chamber with a boat or pontoon. In contrast, we used the static chamber method to measurement the $F$CO$_2$ in small and shallow rivers or streams (<2.5 m wide) which may have caused an overestimation of $CO_2$ evasion to some extent (Lorke et al., 2015). Fortunately, we used the static chamber deployment method only at 4 sites, accounting for about 10% of the total sampling sites. To holistically analyze the $CO_2$ dynamics from headwater small rivers/streams to the downstream large rivers, we have combined the $F$CO$_2$ datasets from deployments of both freely drifting chamber and static chambers. We also realized the potential overestimation from the static chambers, therefore we have discussed the associated uncertainty in the revised manuscript.

**Specific comments**

L 20: I'm not sure that the large uncertainty on the estimate of riverine $CO_2$ emission is due to a lack of data "especially" in headwaters (as stated). There is a generalised lack of high-quality $CO_2$ data everywhere in rivers. Given that 80% of riverine emissions of $CO_2$ are in the tropics, I would assume that largest source of uncertainty is lack of data in tropical areas.

Reply: We completely agree with you. The largest proportion of riverine $CO_2$ emissions are in the tropics and it is critical for us to do more work in these regions. But for future research on the feedback of alpine riverine $CO_2$ emissions to global warming, it is also essential for us to investigate the current riverine $CO_2$ outgassing in this region. The studies on $CO_2$ emissions from the Yellow River were mainly confined to its middle and lower reaches and the estuary. In contrast, little has been done in its upper reaches, especially the source region on the Tibetan Plateau. The Yellow River source region is located in an alpine zone with the mainstream and its tributaries flowing through a variety of land cover types, including grassland, wetland, glacier, and permafrost. Affected by increasing temperature due to global warming, these alpine rivers have become hot spots of riverine carbon cycle studies and warrant a thorough understanding of their implications in the context of global climate change (Ulseth et al., 2018; Peter et al., 2014; Hood et al., 2015). In particular, although Ran et al., (2015a; b) have used compiled water chemistry data to estimate $pCO_2$ and $FCO_2$, there are no field-based direct measurements of $CO_2$ emissions from these alpine rivers.

L 45: Please rephrase. Researchers do not "believe". They build theories and test hypotheses.
Reply: Rephrased.
Now it reads "Many researchers have argued that…".

L46: Emerging evidence (Abril et al. 2014) points to the importance of wetlands in driving riverine $CO_2$ emissions rather than "terrestrial ecosystems", as these are conceptually difference sources of carbon for rivers.
Reply: Thanks for your comments. We have added the related description into the revised manuscript.
Now it reads 'Abril et al. (2014) pointed that wetlands are the primary source of riverine $CO_2$ emissions in the Amazon river…'.

L 55: Lauerwald et al. (2015) gives a global estimate of 0.65 Pg C $yr^{-1}$.
Reply: We have added this result to our revised manuscript.
Now it reads '…recent global $CO_2$ outgassing fluxes from rivers and streams range from 0.65 to 3.2 P g C $yr^{-1}$…'.

L55: The value of 0.7 PgC/yr from Cole et al. (2007) is for all inland waters. Cole et al. (2007) give a value of 0.2 PgC/yr for rivers alone.
Reply: Thanks for your kind reminder. We have corrected the '0.75' to '0.23'.
Now it reads '…higher than the earlier estimate by Cole et al. (2007) (i.e., 0.23 P g C $yr^{-1}$).'.

L 64: studies instead of studied

Reply: Thanks for your suggestions. We have changed 'studied' to 'studies'.
Now it reads '…there are limited studies on CO2 effluxes of rivers in extreme geographical and climatic conditions…'.

L78-79: can you please develop the differences of "underlying control mechanisms" for $CO_2$ emissions between alpine climate and other climates? For me it's the same mechanisms, but it's just colder.

Reply: We agree with you. The lower temperature is one of the characteristics of the alpine climate. In addition, it has many other environmental characteristics, such as stronger sunshine, lower air pressure, and lower precipitation compared with other climates, which are the essential conditions for affecting riverine carbon transport.

Furthermore, export of DOC is important to riverine $CO_2$ outgassing, and the sources of DOC are different. For example, in tropical wetland of the Amazon River system, Mayorga et al. (2005) find that respiration of contemporary organic matter (less than five years old) originating on land and near rivers is the dominant source of $CO_2$ that drives outgassing in medium to large rivers. Abril et al. (2014) further showed that the flooded forests and floating vegetation export large amounts of carbon to river waters and the Amazonian wetlands export half of their gross primary production to river waters as dissolved $CO_2$ and organic carbon. For the Wuding River flowing through arid and semi-arid Loess Plateau in north China, Ran et al. (2017) concluded that lateral carbon derived from soil respiration and chemical weathering played a central role in controlling the variability of riverine $pCO_2$. For the Wuding River, Ran et al. (2018) also showed that enhanced organic matter inputs from agricultural tillage in spring and from terrestrial ecosystems in summer are the major sources of riverine carbon, and radiocarbon analysis suggests the release of old carbon previously stored in soil horizons.

Globally, approximately 13% of the annual flux of glacier dissolved organic carbon is a result of glacier mass loss. These losses are expected to accelerate, leading to a cumulative loss of roughly 15 teragrams (Tg) of glacial dissolved organic carbon by 2050 (Hood et al., 2015). The storage of soil organic carbon in the Arctic and subarctic regions is about 1672 Pg, accounting for about half of the global soil carbon storage (Ping et al., 2008, Tarnocai et al., 2009). And the DOC transported from the permafrost to the Arctic Ocean by large rivers accounts for 11% of the global river DOC flux (Finlay et al., 2006). Unlike other areas, the glacier and frozen soils are important DOC sources in boreal regions which could have strong response to global warming.

L93-102: Ran et al. (2017) reports very extensive data-set in the Wuding River (part of the Yellow river basin) with data obtained at altitudes up to 1340m. So, some information is already available.

Reply: Thanks for your comments. The carbon dynamics in the semi-arid Wuding catchment

has been compared to that in the Yellow River source region. We have added a comparative analysis into the revised manuscript.

Now it reads 'Ran et al. (2017) further studied the Wuding River, a tributary of the middle Yellow River, and concluded that lateral carbon derived from soil respiration and chemical weathering played a central role in controlling the riverine $p\mathrm{CO_2}$. In addition, radiocarbon analyses of the degassed $\mathrm{CO_2}$ suggest the release of old carbon previously stored in soil horizons (Ran et al., 2018).'.

L 155: The determination of end-point with Methyl orange indicator seems a bit crude. Can you please state the estimated accuracy of the method? Did you check the accuracy with standards made of $\mathrm{NaCO_3}$?

Reply: Yes, we have checked the accuracy with $\mathrm{NaCO_3}$ standards before each experiment. Total alkalinity was determined by triplicate titrations in the field with 0.1 M HCl, and methyl orange was used as the indicator, following the standards as suggested by APHA (1999, Standard Methods for the Examination of Water and Wastewater). Our field triplicate titration results are highly consistent with the difference between the three results generally less than 3%. Thus, we expected that the obtained alkalinity results are reliable with high confidence. Finally, DIC was calculated from total alkalinity, pH, and temperature by using the program $\mathrm{CO_2}$calc. Because the measured pH varied from 7.0 to 9.0, the calculated DIC was approximately equal to alkalinity, with >96% of the alkalinity composed of $\mathrm{HCO_3^-}$, consistent with the relative speciation (%) of $\mathrm{CO_2}$, $\mathrm{HCO_3^-}$, and $\mathrm{CO_3^{2-}}$ in water as a function of pH (please refer to Figure 2 below). In fact, the concentration of required HCl we prepared was usually not exactly 0.1M because of manmade errors. In this case, we used $\mathrm{NaNO_3}$ standards to calibrate the HCl concentration prior to titration. Therefore, the actual HCl concentration was usually 0.098 or 0.099 M, and we used this number (0.098 or 0.099), instead of 0.1, for the titration calculation.

[Figure]

Figure 2: Relative concentrations of the different inorganic carbon compounds against pH.

L 245: You cannot conclude that the dampening effect of chambers is responsible for the lack of correlation between wind and K600. If k600 is overwhelmingly driven by other processes than wind, then you would also arrive at a lack of correlation irrespective of a dampening effect.

Reply: There are many factors affecting the gas transfer velocity, e.g., wind speed, flow velocity, depth, slope, discharge etc (Wanninkhof et al., 1992; Zappa et al., 2007; Raymond et

al., 2012). Below are the linear relations between these factors and $K_{600}$.

[Figure]

The results show that, slope and flow velocity show a relatively positive relationship with the $k_{600}$. This means that all these factors did not overwhelm the flow velocity in affecting the gas transfer velocity. Previous studies indicate that there are two main reasons for the overlooking in the relationships between gas transfer and wind speed. Firstly, the short-term monitoring of wind speed is less stable than long-term averaged winds to estimate gas transfer velocities. Another factor frequently overlooked is that the chemical enhancement of $CO_2$ exchange will increase $CO_2$ fluxes at low wind speeds (Wanninkhof et al., 1992). Small-scale waves have been suggested as a dominant mechanism for $k_{600}$ (Bock et al., 1999). Surface contamination by thin organic films measured in the field has also been shown to dampen high frequency waves and leads to reduced gas exchange (Frew et al., 2004). Less dependence of $k$ is observed on wind speed under conditions when buoyancy dominates the production of turbulence in the near-surface layer (McGillis et al., 2004). During very low winds, gas exchange is controlled by tide-driven surface turbulence within the aqueous surface boundary layer in rivers and estuaries. (Zappa et al., 2007). Overall, the dampening effect is likely the

reason for the obscure linear relationship between wind velocity and $k_{600}$.

L 252: clarify statement "mainly due to the water temperature played a crucial role in limiting $CO_2$ transfer between the air-water interface". Since the gas transfer is normalized to Schmidt number of 600, this automatically removes the effect of water temperature.
Reply: Here we argue that a low temperature condition will have two effects on the $CO_2$ emission from river water. Firstly, the low temperature conditions limit the rate of Brownian motion and reduce the exchange of $CO_2$ across the water-air interface. Secondly, low temperatures will increase the solubility of the gas in water, which reduces the outgassing of $CO_2$. A clearer description has been replaced in the manuscript. Now it reads 'A low temperature will limit the rate of Brownian motion and reduce the $CO_2$ exchange with the atmosphere. Meanwhile, a low temperature will increase the solubility of dissolved $CO_2$, thus reducing the outgassing of $CO_2$.'.

L 303: the relationship between $pCO_2$ and DOC can also indicate that both have a common origin such as (simultaneous) inputs from soils. It does not imply that DOC "supports" $CO_2$ production as stated.
Reply: Thanks for your comments. The unappreciated description has been removed because we did not perform related experiments about riverine $CO_2$ source.

L 349: Abril et al. (2014) report on the influence of floodplain lakes on $CO_2$ dynamics in the Amazon rivers, not "peatland rivers" as stated.
Reply: Thanks for your comments. The uncorrected citation has been removed.

L 349: Hu et al. 2015 is missing from the reference list. I assume it's Hu 2005, that seems to correspond to a PhD thesis from one of the co-authors, possibly based on part of the data reported in this paper, so corresponding to circular auto-citation.
Reply: Sorry for having listed the reference Hu (2005) in an incorrect order. The Hu (2005) is a Master thesis work in the same research area and does not belong to any co-authors. The corrected order has been listed in the reference.

L375: rephrase "Although groundwater is participated"
Reply: Rephrased.
Now it reads 'In addition, DIC is an important source of riverine $CO_2$ for grassland rivers. While stream DIC source are highly variable across space and time (Smits et al., 2017), most of the $HCO_3^-$ in the Yellow River source region is derived from carbonate and silicate weathering (Wu et al., 2005; Wu et al., 2008; Wu et al., 2008), which largely reflects the contribution of groundwater inflow (Marx et al., 2017).'.

Legend of figure 2: On what grounds did you exclude k600 data above 70 m/d? It is very awkward to exclude data without justification.
Reply: The large values we excluded are mostly concentrated on the modeled part. There are

a number of factors affecting the $k_{600}$, such as wind speed, slope, flow velocity, depth, and discharge as mentioned above. Thus, using only flow velocity and slope of river channels would have caused overestimation for mountainous rivers due to their relatively high channel slope and thus higher flow velocity. Therefore, we have removed the extremely high $k_{600}$ data points from analysis. We have provided a detailed justification in the revised manuscript, and now it reads 'Using only flow velocity and slope of river channels would have caused overestimation for mountainous rivers due to their relatively high channel slope and thus higher flow velocity. Therefore, the extremely high $k_{600}$ values calculated from Raymond et al. (2012) Equation (18) were excluded from the comparison between our calculated $k_{600}$ and the modeled $k_{600}$.

$$K_{600} = VS \times 2841 \pm 107 + 2.02 \pm 0.209 \tag{18}$$

where, V is the stream velocity (m s$^{-1}$), S is the slope of rivers (unitless).

In figures 5-8 it could be useful to plot $p$CO$_2$ versus water temperature.

Reply: Below is the linear relation between water temperature and $p$CO$_2$.

[Figure]

It is known that water temperature could play an important role in controlling riverine organic matter degradation (Battin et al., 2008), but the analysis in complex river network structures and land cover types (i.e., glacier, permafrost, wetland, and grassland) did not showed a statistically significant linear relationship. Thus, we did not add this figure into our revised manuscript.

**References**

Abril, G., Martinez, J.M., Artigas, L. F., Moreira-Turcq, P., Benedetti, M. F., Vidal, L., Meziane, T., Kim, J.-H., Bernardes, M. C., and Savoye, N.: Amazon River carbon dioxide outgassing fueled by wetlands, Nature, 505, 395–398, 2014.

Battin, T. J., Kaplan, L. A., Findlay, S., Hopkinson, C. S., Marti, E., Packman, A. I., Newbold, J. D., and Sabater, F.: Biophysical controls on organic carbon fluxes in fluvial networks. Nature Geoscience, 1 (8), 95–100, 2008.

Bock, E. J., Hara, T., Frew, N. M., and Mcgillis, W. R.: Relationship between air-sea gas transfer and short wind waves. Journal of Geophysical Research Oceans, 104(C11), 25821-25831, 1999.

Cole, J. J., Prairie, Y. T., Caraco, N. F., McDowell, W. H., Tranvik, L. J., Striegl, R. G., Duarte, C. M., Kortelainen, P., Downing, J. A., Middelburg, J. J., and Melack, J.: Plumbing the global carbon cycle: Integrating inland waters into the terrestrial carbon budget, Ecosystems, 10, 171–184, 2007.

Dickson, A. G., Sabine, C. L., and Christian, J. R.: Guide to best practices for ocean $CO_2$ measurements. Pices Special Publication, 2007.

Finlay, J., Neff, J., Zimov, S., Davydova, A., and Davydov, S.: Snowmelt dominance of dissolved organic carbon in high-latitude watersheds: Implications for characterization and flux of river DOC. Geophysical Research Letters, 33(10): 229-243, 2006.

Frew, N. M., Bock, E. J., Uwe, S., Tetsu, H., Horst, H., Edson, J. B., McGillis, W.R., Nwlson, R. K., McKenna, S. P., Uz, B. M., and Ja¨hne, B.:Air-sea gas transfer: its dependence on wind stress, small-scale roughness, and surface films. Journal of Geophysical Research, 109(8), 371-375, 2004.

Hood, E., Battin, T. J., Fellman, J., O'Neel, S., and Spencer, R. G. M.: Storage and release of organic carbon from glaciers and ice sheets. Nature Geoscience, (2), 59-63, 2015.

Lorke, A., Bodmer, P., Noss, C., Alshboul, Z., Koschorreck, M., Somlai-Haase, C., Bastviken, D., Flury, S., McGinnis, D. F., Maeck, A., Müller, D., and Premke, K.: Technical note: drifting versus anchored flux chambers for measuring greenhouse gas emissions from running waters, Biogeosciences, 12, 7013-7024, 2015.

Lueker T, Dickson A, and Keeling C. Ocean $pCO_2$ calculated from dissolved inorganic carbon, alkalinity, and equations for $K_1$ and $K_2$: Validation based on laboratory measurements of $CO_2$ in gas and seawater at equilibrium. Marine Chemistry, 70(1):105-119, 2000.

Marx A, Dusek J, Jankovec J, Sanda, M., Vogel, T., Geldern, R.V., Hartmann, J., and Barth, J.A.C.: A review of CO2 and associated carbon dynamics in headwater streams: A global perspective. Reviews of Geophysics, 55(2):560-585, 2017.

Mayorga, E., Aufdenkampe, A. K., Masiello, C. A., Krusche, A. V., Hedges, J. I., & Quay, P. D., Richey, J. E., and Brown, T. A.: Young organic matter as a source of carbon dioxide outgassing from amazonian rivers. Nature, 436(7050), 538-541, 2005.

Mcgillis, W. R., Edson, J. B., Zappa, C. J., Ware, J. D., Mckenna, S. P., Terray, E. A., Hare. J. E., Fairall. C. W., Drennan. W., Donelan, M., DeGrandpre, M. D., Wanninkhof, R., and Feely. R.A.: Air-sea co 2 exchange in the equatorial pacific. Journal of Geophysical Research Oceans, 109(8), 371-375, 2015.

Peter, H., Singer, G. A., Preiler, C., Chifflard, P., Steniczka, G., and Battin, T. J.: Scales and drivers of temporal pCO2 dynamics in an Alpine stream, Journal of Geophysical Research: Biogeosciences, 119, 1078–1091, 2014.

Ping, C. L., Michaelson, G. J., Jorgenson, M. T., Kimble, J. M., Epstein, H., Romanovsky, V. E., and Walker. D.A.: High stocks of soil organic carbon in the north american arctic region. Nature Geoscience, 1(9), 615-619, 2008.

Ran, L., Lu, X X., Richey, J E., Sun, H., Han, J., Yu, Y., Liao, S., and Yi, Q.: Long-term spatial and temporal variation of CO2 partial pressure in the Yellow River, China. Biogeosciences, 2015a, 12(4):921-932.

Ran, L., Lu, X. X., Yang, H., Li, L., Yu, R., Sun, H., and Han, J.: CO2 outgassing from the Yellow River network and its implications for riverine carbon cycle. Journal of Geophysical Research: Biogeosciences, 2015b, 120:1334–1347.

Ran, L., Li, L., Tian, M., Yang, X., Yu, R., Zhao, J., Wang, L., Lu, X.: Riverine CO2 emissions in the Wuding River catchment on the Loess Plateau: Environmental controls and dam impoundment impact. 122 (6), 2017.

Ran, L., Tian, M., Fang, N., Wang, S., Lu, X., Yang, X., and Frankie, C.: Riverine carbon export in the arid to semiarid Wuding River catchment on the Chinese Loess Plateau. Journal of Geophysical Research Biogeosciences, 15(12), 3857-3871, 2018.

Raymond, P.A., Zappa. C.J., Butman. D., Bott. T. L., Potter. J., Mulholland. P., Laursen. A. E., McDowell. W. H., and Newbold. D.: Scaling the gas transfer velocity and hydraulic geometry in streams and small rivers. Limnology and Oceanography: Fluids and Environments, 2(1), 2012.

Smits, A. P., Schindler, D. E., Holtgrieve, G. W., Jankowski, K. J., and French, D. W.: Watershed geomorphology interacts with precipitation to influence the magnitude and source of CO2 emissions from Alaskan streams, Journal of Geophysical Research: Biogeosciences, 122, 1903–1921, 2017.

Tarnocai, C., Canadell, J. G., Schuur, E. A. G., Kuhry, P., Mazhitova, G., and Zimov, S.: Soil organic carbon pools in the northern circumpolar permafrost region. Global Biogeochemical Cycles, 23(2), 2009.

Ulseth, A. J., Bertuzzo, E., Singer, G. A., Schelker, J., and Battin, T. J.: Climate-induced changes in spring snowmelt impact ecosystem metabolism and carbon fluxes in an alpine stream network, Ecosystems, 21, 373–390, 2018.

Wanninkhof R.: Relationship between wind speed and gas exchange over the ocean.: Journal of Geophysical Research Oceans, 97(C5):7373-7382, 1992.

Weiss, R. F.: Carbon dioxide in water and seawater: the solubility of a non-ideal gas, Marine Chemistry, 2, 203–215, 1974.

Wu, L., Huh, Y., Qin, J., Gu, D., and Lee, S.: Chemical weathering in the Upper Huang He (Yellow River) draining the eastern Qinghai-Tibet Plateau. Geochimica Et Cosmochimica Acta, 69(22):5279-5294, 2005.

Wu, W., Yang, J., Xu, S., and Yin, H.: Geochemistry of the headwaters of the Yangtze River, Tongtian He and Jinsha Jiang: Silicate weathering and $CO_2$ consumption. Applied Geochemistry, 23(12):3712-3727, 2008.

Wu, W., Xu, S., Yang, J., and Yin, H.: Silicate weathering and $CO_2$, consumption deduced from the seven Chinese rivers originating in the Qinghai-Tibet plateau. Chemical Geology, 249(3), 307–320, 2008.

Zappa, C. J., Mcgillis, W. R., Raymond, P. A., Edson, J. B., Hintsa, E. J., Zemmelink, H. J., Dacey, J.W.R., and Ho. D.T.: Environmental turbulent mixing controls on air-water gas exchange in marine and aquatic systems. Geophysical Research Letters, 34(10), 373-373, 2007.

---

## Author Comment (AC2) · 22 Oct 2018

Dear reviewer,

We are grateful to your comments on the manuscript. Based on your very constructive comments, we have thoroughly revised the manuscript. We have also responded below to all your comments. Please see below the details. Major revisions have also been highlighted in the revised manuscript in green color.

With best regards
Mingyang Tian, on behalf of the coauthors

The manuscript by Mingyang Tian and co-authors investigates the riverine partial pressure of $CO_2$ and $CO_2$ efflux in the source region of the Yellow River and differentiates between landscape types (glacier, permafrost, wetland and grassland). This approach is different than the most studies about $CO_2$ in rivers. Commonly studies compare streams/rivers by size or by climate zone (Marx et al. 2017, Lauerwald et al. 2015, Raymond et al. 2013). Thus, this study aims to improve the understanding of carbon dioxide emissions in alpine rivers, particularly in the Tibetan Plateau. Further, different methods to determine CO2 degassing were applied: floating chambers and headspace equilibrium method. Unfortunately, the uncertainties of results from the different methods are hardly discussed.

Overall, in my opinion the work has a high potential to make a good contribution to the understanding of $CO_2$ emission in rivers. However, at present several aspects require extensive revision. That is why I cannot recomment this manuscript for publication in its present form. I recommend re-submission after thorough revision according to the points below.

Major comments:

(1) This promising paper is restricted to the Yellow River source regions and lacks moving beyond that. I recommend showing results in a global context.

Reply: We have added the discussion of global context into the revised manuscript.

Now it reads 'While the Yellow River source region occupies 17.6% of the whole Yellow River basin, it accounts for only around 4% of the basin's total $CO_2$ efflux (Ran et al., 2015a; 2015b). The $CO_2$ effluxes of the Yellow River source region is also small compared with the efflux from boreal river catchments (Teodoru et al., 2009; Butman and Raymond., 2011; Crawford et al., 2013; 2015; Kokic et al., 2015; Looman et al., 2016) or even smaller relative to the global $CO_2$ efflux (Aufdenkampe et al., 2011).'.

(2) The final version of the paper would benefit from editing for language. While it is generally understandable, several idiomatic expressions and mistakes hamper the readability. For instance, make sure the text is in past-tense and use "the" before plural and delete it if singular follows.

Reply: We have thoroughly polished the language throughout the text. Many thanks for your comments.

(3) I strongly recommend to discuss the reliability of your data. For instance, how were fluxes determined by chamber method compared to fluxes by equilibrium headspace method?

Reply: Thanks for your comments. We have discussed the reliability mainly through three parts. Firstly, we used two methods to evaluate riverine $CO_2$ emissions (i.e., determine $FCO_2$ with the floating chamber method and determine $pCO_2$ with Dickson et al. (2007). Secondly, we discussed the calibrated $pCO_2$ against the $pCO_2$ based on CO2SYM. Finally, we discussed the reliability of two $k_{600}$ datasets (i.e., $k_{600}$ calculated with $pCO_2$ and $FCO_2$ data; $k_{600}$ determined by using the Model 5 of Raymond et al., (2012).

Specific comments:
Abstract
Ln. 1: Don't use the same sentence to start the abstract and the introduction
Reply: This sentence has been rephrased.
Now it reads 'Under the context of climate change, studying $CO_2$ emissions in alpine rivers is important because of the huge carbon storage in these terrestrial ecosystems.'.

Ln. 25: They mostly were a $CO_2$ source. I remember that you also showed negative fluxes.
Reply: This sentence has been reworded.
Now it reads 'The results showed that most of the rivers in the Yellow River source region were a net $CO_2$ source...'.

Ln. 32: Be careful with "significant" as it indicates a statistical significance. Is that the case here? Otherwise replace.
Reply: Replaced with 'considerably'.

Ln. 32-33: Rephrase sentence it is difficult to understand.
Reply: Rephrased.
Now it reads 'Although the rivers in the Yellow River source region annually release little $CO_2$, there is a high carbon evasion potential.'.

Ln. 33-37: These sentences include general knowledge. Give some numbers or statements derived from you research, as this increases the impact of the paper.
Reply: Replaced, please see lines 34 to 37.
Now it reads 'Our study suggested that the dissolved organic carbon (DOC) in permafrost rivers ($5.0\pm2.4$ mg $L^{-1}$) is equivalent to that in peatland covered rivers ($5.1\pm3.7$ mg $L^{-1}$), and the DOC is mainly derived from old carbon stored in frozen soils. In addition, for glacial rivers with limited supply of exogenous carbon, the intensity of $CO_2$ emissions is still considerable. Therefore, with rising temperature due to global warming, increased $CO_2$ emissions in these regions should not be ignored for a better assessment of global riverine $CO_2$ emissions.'.

Introduction

Ln. 45: Researcher should not believe. General agreement/ consensus/ …?

Reply: Replaced with 'Many researchers have argued that'.

Ln. 48-52: Please revise sentence as it is hardly readable.

Reply: Rephrased.

Now it reads 'Therefore, to more accurately estimate riverine $CO_2$ outgassing and understand its driving factors, more studies focusing on rivers in particular climates (i.e., alpine climate) and regions (e.g., headwater region or intermitted rivers) are strongly needed to gain deeper insights into global carbon balance processes.'.

Ln. 55-57: Check Swakuchi et al. 2017. They included lower reaches of the amazon to global estimate by Raymond et al. (2013). This led to values of 2.58 petagrams (Pg) $CO_2$ yr$^{-1}$ for rivers and streams. See also Marx et al. (2017).

Reply: Added. And we have not only added the Swakuchi et al. (2017) result, but also the Lauerwald et al. (2015) into the revised manuscript.

Now it reads 'For example, recent global $CO_2$ outgassing fluxes from rivers and streams range from 0.65 to 3.2 P g C yr$^{-1}$ (Raymond et al., 2013; Lauerwald et al.,2015; Swakuchi et al. 2017; Drake et al., 2017), which are considerably higher than earlier estimate by Cole et al. (2007) (i.e., 0.23 P g C yr$^{-1}$).'

Ln 57-60: This sentence is not clear. Do you mean there is a lack of direct measurement data? That is true. The studies you mention calculate pCO2 from DIC/Alkalinity, pH and T, as this is the common method. There is a decent database (Hartmann et al. 2014), that was basis for global $CO_2$ emission estimates (Raymond et al. 2013; Lauerwald et al. 2015).

Reply: Thanks for your comments.

Now it reads 'A major reason for huge range is because of the absence of a global $CO_2$ outgassing database which includes direct $CO_2$ emission measurements over different rivers…'.

Ln. 61: What do you mean by global river systems?

Reply: To make the statement clearer, we have reworded the statement.

It now reads 'More direct field measurements are therefore strongly needed to better refine global $CO_2$ efflux estimates.'

Ln. 73: Wrong word. This was not "concluded" but determined/measured.

Reply: Thanks for your kind reminding.

We have changed the 'conclude' to 'determined'.

Ln. 86: Statistically significant? Can you give a number?

Reply: We have removed the wording. Not it reads 'significant'.

Ln. 87-88: Do you compare autumn values here?

Reply: Yes, the comparation is conducted by Zhang et al. (2009). We have rephrased the statement.

Now it reads 'Zhang et al. (2009) measured $F$CO$_2$ of the Yellow River and concluded that the Yellow River waters were a source of atmospheric CO$_2$ during autumn and the flux was about 0.0174 Tg C, which was similar to that of the Ottawa River but far less than that of the Amazon in autumn.'.

Ln. 102: In the alpine rivers of the Yellow River?

Reply: The source region is only the alpine part of Yellow River.

Ln. 107-112: Good!

Reply: Thanks for your comments.

Material and methods

Later in the text you mention carbonate rocks/limestone. I recomment to provide data on bedrock geology in this section.

Reply: We have added the related description to revised manuscript.

Now it reads 'Its lithology is homogeneous and predominantly composed of shale and granite rocks (Chen et al., 2005).

Ln. 138: analyses?

Reply: We have changed the 'analysis' to 'analyses'.

Ln. Accuracy of 0.004 for pH measurements in the field. Is that really realistic?

Reply: The accuracy of the WTW pH probe is ±0.004. During our field *in situ* pH measurement of river water, we recorded the pH value when the reading of the pH probe is stable, usually 2 minutes after being put into the river water. The recorded stable pH was used to represent the river water pH.

Ln. 150: This is basic knowledge: inorganic carbon species distribution dependency on pH (and temperature).

Reply: Yes, we consider most of the inorganic carbon as HCO$_3^-$ when calculating the dissolved inorganic carbon species from alkalinity.

Ln. 153: Be consistent and add the country behind the company.

Reply: Added, USA.

The full citation reads 'Whatman GF/F, GE Healthcare Life Sciences, USA'.

Ln. 57: "Specific bottle" be more precise! Brown glass bottle? Volume?

Reply: We have provided a more detailed description of the sampling procedure. They are 100 ml amber glass vials.

Now it reads '…the remaining filtered water was transferred into 100 ml amber glass vials…'.

Ln 159: Germany
Reply: Corrected.

Ln. 160: Not less. "… precision better than …"
Reply: Thanks for your comments, we have changed it to 'precision better than'.

Ln. 162: Use Determination instead of calculation here.
Reply: Thanks for your comments, we have changed it to 'determination'.

Ln. 164-165: Can you give a precision of the Li-7000 analyzer?
Reply: A precision better than 1%, added.
Now it reads '…which has a precision better than 1%...'.

Ln. 175-176: Were 6-10 mins sufficient to get data for a linear flux estimation?
Reply: Yes, the 6-10 minutes of chamber deployment is enough to determine the flux. Below is a screenshot of the slope of $CO_2$ concentration against time. We can see that there is a stable increasing trend of $CO_2$ concentration after a short turbulence. Therefore, we used the this steadily increasing trend as the slope ($R^2$ usually higher than 0.97) to determine the $CO_2$ outgassing flux.

[Figure]

Figure. The screenshot of $CO_2$ concentration accumulation against time.

Ln. 181-190: A better description is needed here. The first equation is a linear approximation of $CO_2$ flux from chamber measurements. The second is mostly applied when $pCO_2$ is calculated from DIC/Alkalinity, pH and T. However, here it can be used to estimate $k_H$.
Reply: Based on your comments, we have reworded the description. Firstly, we used the linear approximation of $CO_2$ flux from chamber measurements to calculate the $CO_2$ outgassing flux. Secondly, we used the Dickon et al., (2017) method to calibrate the headspace-based $pCO_2$. Finally, we utilized the obtained $CO_2$ outgassing flux (i.e., $FCO_2$) and river water $pCO_2$ to determine $K_{600}$.

Ln. 200: "… blow±3
Reply: Thanks for your comments, we have corrected it to 'analytical error below ±3%'.

Ln. 195-208: Good method. Did you ever apply chambers until equilibrium was reached? It would be interesting to see if results match.

Reply: Unfortunately, we did not apply chambers until equilibrium was reached because it would need around 1 hour for every chamber deployment. In our future field measurements, we will try to determine the river water $p\text{CO}_2$ by waiting for the equilibrium and then compare the two methods. Many thanks for your suggestions.

Results

Ln. 218-219: Please add the year.

Reply: We have added the year.

Now it reads 'The annual average air temperature in 2016 was 16.7±6.3 °C.'.

Ln. 229: The last part of the sentence is interpretation and belongs to the discussion section.

Reply: Thanks for your comments, we have removed this sentence.

Ln. 244: This is new. You did not mention with in the MaterialMethods section! I thought you determined k600 by rearranging equation (2)? This belongs to the MaterialMethods section.

Reply: Thanks for your comments. We have added the description of citation of Raymond et al. (2012) method to the Section Material and Methods. In addition, we have moved this discussion part to the discussion section.

Ln. 245: Computed? From what? Chamber measurements? I guess with model results you refer to model 5 by Raymond et al. (2012).

Reply: Thanks for your comments. We have added the material methods about this.

Now it reads 'We also predicting the $K_{600}$ (m d$^{-1}$) through the Model 5 given by Raymond et al. (2012).

$$K_{600} = VS \times 2841 \pm 107 + 2.02 \pm 0.209$$

Where, V is the stream velocity (m s$^{-1}$), S is the slope of rivers (unitless).

In addition, the large values we excluded are mostly concentrated on the modeled part. There are many factors affecting the K600, such as wind speed, slope, flow velocity, depth, and discharge as mentioned above. Thus, using only flow velocity and slope of river channels would have caused overestimation for mountainous rivers due to their relatively high channel slope and thus higher flow velocity. Therefore, we have removed the extremely high $K_{600}$ data points from analysis.'

Ln. 246: This belongs to the discussion, not results.

Reply: Thanks for your comments, we have moved this section to the discussion section.

Now it reads 'With respect to the $k_{600}$, the computed $k_{600}$ showed statistically significant but weak correlation with the modeled results when the high $k_{600}$ values (>70 m d$^{-1}$) were removed from analysis. Given the chamber's dampening effect of wind (Matthews et al., 2003), there was no any statistically significant relationship between wind and $k_{600}$ for streams. Instead, flow

velocity is a relatively good predictor of $k_{600}$ and can approximately explain 15% of its variability. Although we deployed the floating chamber very carefully, the statistical analysis could not reflect the complex interactions of various environment factors except the four land cover types through our 36 sampling sites. Additionally, it is worth noting that the Model 5 of Raymond et al. (2012) has overestimated the $k_{600}$, especially for mountainous rivers. This is probably because of low water temperature that has constrained $CO_2$ degassing although the steeper channel slope has caused stronger flow turbulence (Battin et al., 2008). A low temperature will limit the rate of Brownian motion and reduce the $CO_2$ exchange with the atmosphere. Meanwhile, a low temperature will increase the solubility of dissolved $CO_2$, thus reducing the outgassing of $CO_2$.'.

Ln. 249-251: This sentence is not understandable. Please rephrase. Be more precise: did you apply a statistical approach to determine relationships?
Reply: Yes, we did a linear statistical analysis of flow velocity and $K_{600}$, and conclude that it could represented 15% of $K_{600}$ ($r^2$=0.15).

Ln. 251-255: This belongs to the discussion, not results.
Reply: Thanks for your comments. We have moved this section to the discussion section.

Ln. 279: Statistically significant? Otherwise don't use "significant".
Reply: Thanks for your comments. We have removed this unappreciated word.

Discussion
Ln. 294: This is a poor beginning. Better describe the key result in a larger context "This study shows/demonstrates…" to create a red line for the forthcoming discussion.
Reply: Corrected.
It now reads 'This study shows that the lowest $FCO_2$ appeared in the permafrost covered region…'.

Ln. 295: Be aware that not all the riverine $CO_2$ is derived from land. Your statement is not correct.
Reply: Yes, the aquatic plants and glacier water or groundwater support amount of river $CO_2$ in alpine river. Corrected.
It now reads 'It is well known that a large quantity of riverine $CO_2$ is derived from land…'.

Ln. 310-312: Please revise this sentence.
Reply: Revised.
Now it reads 'One potential explanation is that its low temperature (i.e., annual average water temperature: 9.9 ℃) because of high elevation may have constrained soil respiration and riverine organic matter degradation.'.

Ln.314: Replace "not easy".

Reply: Replaced.
Now it reads 'it may be difficult for $CO_2$ to degas …'.

Ln. 325-326: This sentence is vague and insignificant. Rephrase with details.
Reply: Rephrased.
Now it reads 'Wang (1998) discovered that these rivers are predominantly supplied by glacier melting that is characterized by significant seasonal variability.'.

Ln. 326: Use past-tense.
Reply: Rephrased.
Now it reads 'The sampled glacier rivers showed the lowest annual average DOC concentration…'.

Ln. 327-329: Rephrase sentence to increase readability.
Reply: Rephrased.
Now it reads 'This is probably because the sub-catchments around the Aemye Ma-chhen Range do not have sufficient vegetation coverage as a result of high elevation and low temperature, limiting the terrestrial source of DOC.'.

Ln. 335: "highlights"
Reply: Rephrased.
Now it reads 'This suggests that the…'.

Ln. 341-344: Rephrase this sentence.
Reply: Rephrased.
Now it reads 'Our observations found that, with increasing distance from the glaciers, the riverine $pCO_2$ exhibited a decreasing trend, which is likely caused by the dilution of glacier-related $pCO_2$.'.

Ln. 346-…: Use past-tense.
Reply: Corrected.
Now it reads 'The river $FCO_2$ was the highest in the peatland coverage area…'.

Ln. 355: Sufficient for what?
Reply: Rephrased.
Now it reads '…which can provide enough riverine $CO_2$.'.

Ln. 366: "Analyses"
Reply: Thanks for your comments. We have changed 'analysis' to 'analyses'.

Ln. 367-368: The relationship between pCO2 and pH is well known. This sentence is not correct, as the pH (and T) determines the species of inorganic carbon in water.

Reply: Thanks for your comments. We have corrected the unappreciated description. Now it reads 'This also shows that $p$CO$_2$ is partially affected by the water pH.'.

Ln. 375-381: What makes you think that groundwater input is higher in grassland regions? If you give a statement like this, you need references. Groundwater samples are not sufficient evidence for this statement, as groundwater $p$CO$_2$ typically are higher than stream $p$CO$_2$.
Reply: Rephrased.
Now it reads 'While stream DIC source are highly variable across space and time (Smits et al., 2017), most of the HCO$_3^-$ in the Yellow River source region is derived from carbonate and silicate weathering (Wu et al., 2005; Wu et al., 2008; Wu et al., 2008), which largely reflects the contribution of groundwater inflow (Marx et al., 2017).'.

Ln. 384: see comment Ln. 294:
Reply: Rephrased.
Now it reads 'This study demonstrates that the annual average $p$CO$_2$ is…'.

Ln. 401-403: There are several other potential reasons. How about pH changes (higher pH means less carbon in the form of CO2)(Stets et al. 2017)? And how about decreasing proportion of groundwater distribution with increasing stream/river size (Marx et al. 2017)?
Reply: Thanks for your comments. We have added these potential reasons into the manuscript. It is common that a higher pH suggests a lower proportion of dissolved CO$_2$ in the DIC species. There is usually an overestimation when using the pH-Alk system to determine the $p$CO$_2$ (Abril et al., 2015). The pH values in our study area ranges from 7.89 to 9.02. Therefore, it should not be the reason for the underestimated $p$CO$_2$.

Ln. 411: "Easily neglected"?
Reply: Rephrased.
Now it reads 'Obviously, it is considerably challenging to detect the impact of groundwater inflow without high-resolution sampling.'

Conclusions
Ln. 445: Revise sentence. Verb missing?
Reply: Revised.
Now it reads 'In the permafrost region, the large amounts of terrestrially-derived DOC supported its high $p$CO$_2$ levels. While in the glacier region, the glacial DOC and CO$_2$ may have played an essential role in determining CO$_2$ outgassing. In the peatland and grassland regions, decomposition of plant-derived organic matter is an important source of riverine CO$_2$. Finally, the ground water and chemical weathering are also played an important role in supporting riverine CO$_2$ in the whole Yellow River source region.'.

Ln. 451: Is this flux for the study area or the whole Yellow River?
Reply: Yes, corrected.

Now it reads 'the riverine CO2 efflux of the Yellow River source region was estimated…'.

Ln. 452: What is the number in Ran et al.? Is the number for the whole Yellow River?
Reply: The number determined by Ran et al., (2015 a, b) was $-0.168 \pm 0.084$ T g C yr$^{-1}$. It is the estimated flux for the source region of the Yellow River only.

Table
Add the year somewhere. The table has to be understandable for itself. Revise subscripts and superscripts, as there are many mistakes.
Reply: We have added the year and corrected the subscripts and superscripts. Please see Table 1 for the additions.

Figure
Figure 1: Good!
Reply: Thanks for your comment.

Figure 2: (a) Are these mean values for all your data? Add a small paragraph in the Material Methods section where you explain your values? Use same names in the Figure title than for the axes labels. Add reference for modeled k600 (Raymond?).
Reply: We have rephrased the title and added a reference of Raymond et al. (2012) into the revised manuscript.

(b)Which k600 did you display here? Measured or modeled ones? Please clearify. The last sentence in your Figure title should be explained in the MaterialMethods section.
Reply: It is the $K_{600}$ measurements based on in situ $p$CO$_2$ and $F$CO$_2$. We also add some explanation into material method section.

Figure 3: Add the dimension for fluxes. What for is the box (red, green) under the legend? Delete?
Reply: Delated the box and the legend are easier to understand.

Figure 4: Write "Figure" with a capital letter at the beginning. Add the year in the Figure title.
Reply: Corrected and added the year.

Figure 5 and 6: Be consistent with brackets: Dimensions either inside or outside brackets.
Reply: Corrected, thanks for your comments.

Figure 7: (d) the a-axis label is not correct.
Reply: Corrected, thanks for your comments.

**References**
Abril, G., Bouillon, S., Darchambeau, F., Teodoru, C. R., Marwick, T. R., Tamooh, F.,

Omengo, F. O., Geeraert. N., Deirmendjian. L., Polsenaere, P., and Borges. A. V.: Technical Note: Large overestimation of $p$CO$_2$ calculated from pH and alkalinity in acidic, organic-rich freshwaters. Biogeosciences, 12, 1(2015-01-06), 12(1): 67-78, 2015.

Aufdenkampe, A. K., Mayorga, E., Raymond, P. A., Melack, J. M., Doney, S. C., Alin, S. R., Aalto, R. E., and Yoo, K.: Riverine coupling of biogeochemical cycles between land, oceans, and atmosphere. Frontiers in Ecology & the Environment, 9 (1), 53–60, 2011.

Battin, T. J., Kaplan, L. A., Findlay, S., Hopkinson, C. S., Marti, E., Packman, A. I., Newbold, J. D., and Sabater, F.: Biophysical controls on organic carbon fluxes in fluvial networks. Nature Geoscience, 1 (8), 95–100, 2008.

Butman, D., and Raymond, P. A.: Significant efflux of carbon dioxide from streams and rivers in the United States, Nature Geoscience, 4, 839–842, 2011.

Chen, J., Wang, F., Meybeck, M., He, D., Xia, X., and Zhang, L.: Spatial and temporal analysis of water chemistry records (1958–2000) in the Huanghe (Yellow River) basin. Global Biogeochemical Cycles, 19(3), 2005.

Cole, J. J., Prairie, Y. T., Caraco, N. F., McDowell, W. H., Tranvik, L. J., Striegl, R. G., Duarte, C. M., Kortelainen, P., Downing, J. A., Middelburg, J. J., and Melack, J.: Plumbing the global carbon cycle: Integrating inland waters into the terrestrial carbon budget, Ecosystems, 10, 171–184, 2007.

Crawford, J. T., Striegl, R. G., Wickland, K. P., Dornblaser, M. M., and Stanley, E. H.: Emissions of carbon dioxide and methane from a headwater stream network of interior Alaska. Journal of Geophysical Research Biogeosciences, 118 (2), 482–494, 2013.

Crawford, J T., Dornblaser, M M., Stanley, E H., Clow D W., and Striegl R D.: Source limitation of carbon gas emissions in high-elevation mountain streams and lakes. Journal of Geophysical Research Biogeosciences, 120 (5): 952–964, 2015.

Dickson, A. G., Sabine, C. L., and Christian, J. R.: Guide to best practices for ocean CO2 measurements. Pices Special Publication, 2007.

Drake, T. W., Raymond, P. A., and Spencer, R. G.: Terrestrial carbon inputs to inland waters: A current synthesis of estimates and uncertainty, Limnology and Oceanography Letters, doi: 10.1002/lol2.10055, 2017.

Kokic, J., Wallin, M. B., Chmiel, H. E., Denfeld, B. A., and Sobek, S.: Carbon dioxide evasion from headwater systems strongly contributes to the total export of carbon from a small boreal lake catchment. Journal of Geophysical Research, Biogeosciences, 120, 13–28, 2015.

Lauerwald, R., G. G. Laruelle, J. Hartmann, P. Ciais, and P. A. G. Regnier.: Spatial patterns in CO$_2$ evasion from the global river network, Global Biogeochem. Cycles, 29, 534–554, 2015.

Looman, A., Santos, I. R., Tait, D. R., Webb, J. R., Sullivan, C. A., and Maher, D.T.: Carbon cycling and exports over diel and flood-recovery timescales in a subtropical rainforest headwater stream. Science of the Total Environment, 550, 645–657, 2016.

Marx, A., J. Dusek, J. Jankovec, M. Sanda, T. Vogel, R. van Geldern, J. Hartmann, and J. A. C. Barth.: A review of CO2 and associated carbon dynamics in headwater streams: A global perspective. Reviews of Geophysics, 55(2):560-585, 2017.

Matthews, C. J., St Louis, V. L., and Hesslein, R. H.: Comparison of three techniques used to measure diffusive gas exchange from sheltered aquatic surfaces. Environmental Science & Technology, 37(4), 772, 2003.

Ran, L., Lu, X X., Richey, J E., Sun, H., Han, J., Yu, Y., Liao, S., and Yi, Q.: Long-term spatial and temporal variation of $CO_2$ partial pressure in the Yellow River, China. Biogeosciences, 2015a, 12(4):921-932.

Ran,L., Lu, X. X., Yang, H., Li, L., Yu, R., Sun, H., and Han, J.: CO2 outgassing from the Yellow River network and its implications for riverine carbon cycle. Journal of Geophysical Research: Biogeosciences, 2015b, 120:1334–1347.

Raymond, P.A., Zappa. C.J., Butman. D., Bott. T. L., Potter. J., Mulholland. P., Laursen. A. E., McDowell. W. H., and Newbold. D.: Scaling the gas transfer velocity and hydraulic geometry in streams and small rivers. Limnology and Oceanography: Fluids and Environments, 2(1), 2012.

Raymond, P. A., Hartmann, J., Lauerwald, R., Sobek, S., McDonald, C., Hoover, M., Butman, D., Striegl, R., Mayorga, E., and Humborg, C.: Global carbon dioxide emissions from inland waters, Nature, 503, 355–359, 2013.

Sawakuchi, H. O., Neu. V., Ward, N.D., Barros. M., Valerio. A., Gagne-Maynard. W., Cunha. A., Less. D., Diniz. J., Brito. C., Krusche. A., and Richey. J.: Carbon dioxide emissions along the lower Amazon River, Front. Mar. Sci., 4(76), 2017.

Smits, A. P., Schindler, D. E., Holtgrieve, G. W., Jankowski, K. J., and French, D. W.: Watershed geomorphology interacts with precipitation to influence the magnitude and source of $CO_2$ emissions from Alaskan streams, Journal of Geophysical Research: Biogeosciences, 122, 1903–1921, 2017.

Teodoru, C. R., del Giorgio, P. A., Prairie, Y. T., and Camire, M.: Patterns in pCO2 in boreal streams and rivers of northern Quebec, Canada. Global Biogeochemical Cycles, 23, 2009.

Wang, J.T.: Climatic Geomorphology of the Anyemaqen Mountains. Journal of Glaciology and Geocryology., 10(2):161-171,1988.

Wu, L., Huh, Y., Qin, J., Gu, D., and Lee, S.: Chemical weathering in the Upper Huang He (Yellow River) draining the eastern Qinghai-Tibet Plateau. Geochimica Et Cosmochimica Acta, 69(22):5279-5294, 2005.

Wu, W., Yang, J., Xu, S., and Yin, H.: Geochemistry of the headwaters of the Yangtze River, Tongtian He and Jinsha Jiang: Silicate weathering and CO consumption. Applied Geochemistry, 23(12):3712-3727, 2008.

Wu, W., Xu, S., Yang, J., and Yin, H.: Silicate weathering and CO2, consumption deduced from the seven Chinese rivers originating in the Qinghai-Tibet plateau. Chemical Geology, 249(3), 307–320, 2008.

Zhang, L., Xu, X., and Wen, Z.: Control factors of $pCO_2$ and $CO_2$ degassing fluxes from the Yellow River in autumn. Advances in Water Science, (2): 227–235, 2009.

---

## Author Comment (AC3) · 22 Oct 2018

The comment was uploaded in the form of a supplement:
https://www.biogeosciences-discuss.net/bg-2018-292/bg-2018-292-AC3-supplement.pdf

---

## Author Comment (AC4) · 22 Oct 2018

**High Riverine CO₂ Outgassing affected by Land Cover Types in the Yellow River Source Region**

Mingyang Tian[1], Xiankun Yang[2], Lishan Ran[3], Yuanrong Su[1], Lingyu Li[1],

Ruihong Yu[1], Haizhu Hu[1*], Xi Xi Lu[1,4*]

[1]Inner Mongolia key laboratory of river and lake ecology, School of ecology and environment, Inner Mongolia University, Hohhot, 010021, China

[2]School of Geographical Sciences, Guangzhou University, Guangzhou, 510006, China

[3]Department of Geography, The University of Hong Kong, Hong Kong, China

[4]Department of Geography, National University of Singapore, 117570, Singapore

✉ *Corresponding author*

Xi Xi Lu Tel.: +86-471-4991469, Fax: +86-471-4991436, e-mail: geoluxx@nus.edu.sg

Haizhu Hu Tel.: +86-471-4991469, Fax: +86-471-4991436, e-mail: huhaizhu@163.com

**Abstract:** Under the context of climate change, studying $CO_2$ emissions in alpine rivers is important because of the huge carbon storage in these terrestrial ecosystems. However, estimates of global riverine $CO_2$ emissions remain highly uncertain owing to absence of a comprehensive $CO_2$ emission measurement, especially in river source regions. In this study, riverine partial pressure of $CO_2$ ($pCO_2$) and $CO_2$ efflux ($FCO_2$) in the Yellow River source region under different landcover types, including glaciers, permafrost, wetlands, and grasslands, were investigated in April, June, August, and October 2016. Relevant chemical and environmental parameters were analyzed to explore the main controlling factors. The results showed that most of the rivers in the Yellow River source region were a net $CO_2$ source, with the $pCO_2$ ranging from 181 to 2441 µatm and the $FCO_2$ from -221 to 6892 g C $m^{-2}$ $yr^{-1}$. Both $pCO_2$ and $FCO_2$ showed strong spatial and temporal variations. Average $FCO_2$ in August was higher than that in other months, with the lowest in October. In alpine climates, low temperature conditions played a crucial role in limiting biological activity and reducing $CO_2$ emissions. The lowest $FCO_2$ values (-221 g C $m^{-2}$ $yr^{-1}$) were observed in the glacier and permafrost regions. By integrating seasonal changes of water surface area, total $CO_2$ efflux was estimated at $0.37\pm0.49$ Tg C $yr^{-1}$, which is considerably higher than previous studies. Although the rivers in the Yellow River source region annually release little $CO_2$, there is a high carbon evasion potential. Our study suggested that the dissolved organic carbon (DOC) in permafrost rivers ($5.0\pm2.4$ mg $L^{-1}$) is equivalent to that in peatland covered rivers ($5.1\pm3.7$ mg $L^{-1}$), and the DOC is mainly derived from old carbon stored in frozen soils. In addition, for glacial rivers with limited supply of exogenous carbon, the intensity of $CO_2$ emissions is still considerable. Therefore, with rising temperature due to global warming, increased $CO_2$ emissions in these regions should not be ignored for a better assessment of global riverine $CO_2$ emissions.

**Key words:** $pCO_2$, $CO_2$ outgassing; glaciers; permafrost; wetland; grassland; Yellow River source region

**1. Introduction**

Rivers connect land and oceans, acting as pipes and containers transporting carbon and other substances from terrestrial ecosystems to the oceans. Existing studies on riverine $CO_2$ evasion focus mainly on the spatial and temporal dynamics of partial pressure of $CO_2$ ($pCO_2$) and $CO_2$ efflux ($FCO_2$) (Cole et al.,2001; Aufdenkampe et al., 2011; Raymond et al., 2013; Abril et al., 2014). Many researchers have argued that

river water $CO_2$ is primarily derived from respiration of terrestrial ecosystems and decomposition of organic matter in river (Raymond et al., 2013; Hotchkiss et al., 2015; Schelker et al., 2016; Ran et al., 2017). For example, Abril et al. (2014) pointed that wetlands are the primary source of riverine $CO_2$ emissions in the Amazon river. However, the sources and underlying mechanisms of riverine $CO_2$ dynamic for many rivers remain largely unknown. Therefore, to more accurately estimate riverine $CO_2$ outgassing and understand its driving factors, more studies focusing on rivers in particular climates (i.e., alpine climate) and regions (e.g., headwater region or intermitted rivers) are strongly needed to gain deeper insights into global carbon balance processes.

With respect to global-scale $CO_2$ outgassing, available estimates are characterized by great uncertainty. For example, recent global $CO_2$ outgassing fluxes from rivers and streams range from 0.65 to 3.2 P g C $yr^{-1}$ (Raymond et al., 2013; Lauerwald et al., 2015; Swakuchi et al., 2017; Drake et al., 2017), which are considerably higher than the earlier estimate by Cole et al. (2007) (i.e., 0.23 P g C $yr^{-1}$). A major reason for the huge range is likely the absence of a global $CO_2$ outgassing database which includes direct $CO_2$ emission measurements over different rivers and under different climate and land cover types (Raymond et al., 2013; Cole et al., 2007; Aufdenkampe et al., 2011; Drake et al., 2017). More direct field measurements are therefore strongly needed to better refine global $CO_2$ efflux estimates.

Yet, there have been few studies on $CO_2$ effluxes of rivers in extreme geographical and climatic conditions, such as alpine rivers (Wu et al., 2008; Zhang et al., 2013). Crawford et al. (2013) investigated the riverine $CO_2$ outgassing in the Alaska region and explored its temporal and spatial changes under different land use types. Crawford et al. (2015) further studied carbon emissions from the rivers and lakes in alpine areas around the Estes Park in the United States and found that the average $pCO_2$ was only 417 μatm. They concluded that high altitude and low vegetation coverage are the primary factors limiting $CO_2$ outgassing. Weyhenmeyer et al. (2015) concluded that production of $CO_2$ in lakes was usually half of the $CO_2$ emissions and most of the degassed $CO_2$ was derived from dissolved inorganic carbon (DIC). Humborg et al. (2010) surveyed rivers in central and northern Sweden and determined that the average $pCO_2$ and $FCO_2$ was 1445 μatm and 3033 g C $m^{-2}$ $yr^{-1}$, respectively. Overall, compared with temperate

and tropical rivers, riverine $CO_2$ outgassing under alpine climate is at a relatively low level. This is largely due to the cold climate with low temperature and high altitude that hamper riverine $CO_2$ emissions (Peter et al., 2014).

75

The riverine $CO_2$ emissions from the Yellow River Basin have been preliminarily studied. Su et al. (2005) reported that the mainstream $pCO_2$ was between 1100 and 1700 µatm, which were in intermediate-low level of world rivers. The main controlling factor was its carbonate system. Zhang et al. (2008) measured the $pCO_2$ of 1570 µatm at Lijin Hydrological Station on the lower Yellow River during sediment

80 regulation period (June–July), which was higher than in other periods. Zhang et al. (2009) measured the $FCO_2$ of the Yellow River and concluded that the Yellow River waters were a source of atmospheric $CO_2$ during autumn and the flux was about 0.0174 Tg C, which was similar to that of the Ottawa River but far less than that of the Amazon in autumn. Ran et al. (2015b) estimated that the annual $CO_2$ emissions of the whole Yellow River system at 7.9 Tg C, which is close to the basin-wide carbon deposition of 8.7 Tg

85 C while larger than the marine import (i.e., 6 Tg C). Ran et al. (2017) further studied the Wuding River, a tributary of the middle Yellow River, and concluded that lateral carbon derived from soil respiration and chemical weathering played a central role in controlling the riverine $pCO_2$. In addition, radiocarbon analyses of the degassed $CO_2$ suggest the release of old carbon previously stored in soil horizons (Ran et al., 2018).

90

These studies on $CO_2$ emissions from the Yellow River were mainly confined to its middle and lower reaches. In contrast, to date little has been done on the upper reaches, especially the source region on the Tibetan Plateau. The Yellow River source region is located in the alpine zone with the Yellow River mainstream flowing through a variety of land cover types, including grassland, wetland, glacier, and

95 permafrost. Affected by increasing temperature as a result of global warming, the alpine rivers in this region have become hot spots of riverine carbon cycle studies and warrant a thorough understanding of their implications for global climate change (Ulseth et al., 2018; Peter et al., 2014; Hood et al., 2015). Although Ran et al. (2015b) have estimated its $pCO_2$ and $FCO_2$ by using water chemistry data, there are no field-based direct measurements of $CO_2$ emissions from these alpine rivers.

To accurately determine the magnitude of riverine $CO_2$ outgassing and understand its underlying control mechanisms in this alpine climate region, we conducted *in situ* measurements of riverine $CO_2$ emissions under different land cover types, including grassland, peatland, glacier, and permafrost, in the Yellow River source region. The objectives of this study were to examine (1) the spatiotemporal patterns of $CO_2$ emissions under different land cover types; (2) the magnitudes of stream $CO_2$ emissions; and (3) the sources of riverine $CO_2$ in this alpine river system. Clearly, the obtained findings will lead to a greater understanding of riverine carbon export and $CO_2$ emissions, especially for alpine rivers, which will help refine the global estimates of riverine $FCO_2$.

**2. Materials and methods**

**2.1 Site description**

The Yellow River originates from the Bayanhar Mountains in Tibetan Plateau, flows through the Loess Plateau and North China Plain, and eventually empties into Bohai Sea. Generally, the drainage basin above the Tangnaihai hydrological station is called the Yellow River source region (Figure1). The study area is situated from 32°3'N 95°5'E to 36°1'N 103°3'E (Figure 1). In this region, most of the rivers flow through the Tibetan Plateau at an altitude of 3000–4000 m with meandering river channels. The study area is about $1.32 \times 10^5$ km², accounting for about 17.6 % of the Yellow River basin. The Yellow River source region is located in an alpine zone with a typical plateau continental climate affected by plateau monsoon (Yang et al.,1991). Its lithology is homogeneous and predominantly composed of shale and granite rocks (Chen et al., 2005). The climate is characterized by a pronounced seasonal variation with the wet season starting from June to September and the dry season from October to next May. Major land cover types of the source region include glacier, permafrost, wetland, and grassland.

Precipitation is the dominant source of runoff in the Yellow River source region. Its annual mean precipitation is 486 mm, accounting for approximately 96% of the total runoff (Liu et al., 2005). The annual evaporation varies from 800 to 1200 mm. Although the area of the source region represents only 17.6% of the whole Yellow River basin, it supplies over 33% of the basin's total water discharge (Sun et

al., 2009). In recent decades, precipitation in the source area has slightly increased owing to accelerating glacier melting (Chang et al., 2007), which has increased its relative importance of water flux for the whole Yellow River basin (Zhang et al., 2012).

**2.2 Fieldwork and laboratory analyses**

In this study, four fieldwork campaigns in the Yellow River source region were conducted in April, June, August, and October 2016. The riverine $pCO_2$ and related environmental factors, including water temperature, pH, dissolved oxygen (DO), were monitored in the field under different land cover types. In total, there are 36 sampling points (Figure 1) and they can be categorized on the basis of complexity of river network structure and land cover types (i.e., glacier, permafrost, wetland, and grassland) (Table 1). In addition, three groundwater samples in grassland covered sub-catchments were collected to determine the $pCO_2$ in groundwater. The temperature, pH, and DO were measured by using a Multi 3420 analyzer (WTW GmbH, Germany) with the accuracies of $\pm0.2$ °C, $\pm0.004$, and $\pm1.5\%$, respectively. Before measurement, the pH probe was calibrated with three pH buffers (i.e., pH4.01, pH7.00, and pH10.01, respectively).

Prior studies suggested that, when pH ranges from 7 to 10, $HCO_3^-$ represents 96% of alkalinity and alkalinity can be used to calculate DIC (Hunt et al., 2011). Alkalinity was determined by on-site titration in this study. The collected water samples were subjected to low-pressure suction filtration through a pre-fired glass fiber filter (Whatman GF/F, GE Healthcare Life Sciences, USA) with a pore diameter of 0.7 μm. For each water sample, the alkalinity was titrated with 0.1 mol L$^{-1}$ HCl within 12 hours after sampling. Triplicate titrations with Methyl orange as the indicator suggest that the analytical error below 3%. Beside alkalinity analysis, the remaining filtered water was transferred into 100 ml amber glass vials, poisoned with nitric acid, and preserved in refrigerator at 4 °C condition for dissolved organic carbon (DOC) measurement in laboratory. DOC was analyzed using a total organic carbon (TOC) analyzer (Elementar Analysensysteme GmbH, Germany), which has a precision better than 3%.

**2.3 Determination of $CO_2$ emission**

The $CO_2$ emission flux $FCO_2$ was measured using the floating chamber method (Ran et al., 2017) with a Li-7000 $CO_2$/$H_2O$ gas analyzer (Li-Cor, Inc, USA), which has a precision better than 1%. The Li-7000 gas analyzer was calibrated with standard $CO_2$ gases of 500 ppm and 2000 ppm before each measurement. The rectangular floating chamber has a volume of 17.8 L and a water surface area of 0.09 $m^2$. The chamber walls were lowered 3 cm into water and mounted with plastic foams that had streamlined ends to limit artificial disruptions to near-surface turbulence. The chamber is covered with tin foil to reduce the influence of sun light's heating. Temperature inside chamber was measured with a waterproof thermometer. Prior to each deployment, the chamber was placed in air and the air inside the chamber was continuously circulated in a closed loop that was connected to the infrared Li-7000 gas analyzer through rubber-polymer tubes. The instrument automatically records the air $CO_2$ concentration and ambient atmospheric pressure. When the chamber was placed on water surface, the accumulating $CO_2$ concentration inside the chamber was recorded every 2 seconds, and each deployment lasted for 6–10 mins. In large rivers with relatively favorable flow conditions, the chamber was tied to a small rubber boat and freely drifted with flow to measure $FCO_2$. In contrast, we used the static chamber method to measure $FCO_2$ in small rivers or streams which may have caused an overestimation of $CO_2$ evasion (Lorke et al., 2015). While the chamber was freely drifting at 32 sampling sites, we used the static deployment method only at 4 sampling sites, accounting for about 10% of the all sites.

The $CO_2$ efflux from water was calculated using following equation (Frankignoulle et al., 1988):

$$FCO_2 = 1000 \times (dpCO_2/dt)(V/RTS) \tag{1}$$

where, $dpCO_2/dt$ is the slope of $CO_2$ change within the chamber (Pa $d^{-1}$; converted from $\mu$atm $min^{-1}$), $V$ is the chamber volume (17.8 L), $R$ is the gas constant, $T$ is the chamber temperature (K), and $S$ is the area of the chamber covering the water surface (0.09 $m^2$ in this study).

Surface water $pCO_2$ was calculated using the headspace equilibrium method (Ran et al., 2017). By using an 1100 mL conical flask, 800 mL of water were collected 10 cm below water surface and the remaining volume of 300 mL was filled with ambient air. The flask was immediately closed with a lid and

vigorously shaken for 1 min to equilibrate the gas in water and air. The equilibrated gas was then injected into the calibrated Li-7000 gas analyzer. Triplicate measurements were performed at each site and the average was calculated (analytical error below ±3%). Surface water $pCO_2$ was calculated based on the equations from Dickson et al. (2007):

$$pCO_2^{water,i} = pCO_2^{headspace,f} + \frac{Vh}{Vw}(pCO_2^{headspace,f} - pCO_2^{headspace,i})/K_0[1 + \frac{K_1}{[H^+]} + \frac{K_1 \cdot K_2}{[H^+]^2}]RT \quad (2)$$

where, the superscripts $i$ and $f$ represent the initial and final $pCO_2$ (μatm), $Vh$ and $Vw$ are the headspace volume and water volume, respectively, $K_0$ is the solubility of $CO_2$ in water calculated on the basis of solubility constants for $CO_2$ from *Weiss* (1974), $K_1$ and $K_2$ are the thermodynamic reaction constants (Lueker et al., 2000), [$H^+$] represents the total concentration of hydrogen ions in final solution. $R$ is the universal gas constant (8.314 J mol$^{-1}$ K$^{-1}$), and $T$ is the water temperature (K). Temperature in the flask after equilibration was measured to correct for temperature changes relative to that of in situ river water. The initial $pCO_2$ was taken as the $CO_2$ concentration in ambient air before the headspace equilibration measurement.

Conventionally, $FCO_2$ can also be estimated from the following equation.

$$FCO_2 = k \cdot K_H \cdot \Delta pCO_2 \quad (3)$$

where, $k$ is the gas transfer velocity (m d$^{-1}$), $K_H$ is the Henry's constant for $CO_2$ at a given temperature, $FCO_2$ is the measured riverine $CO_2$ efflux, and the $\Delta pCO_2$ is the difference between the surface water and the atmosphere. Using the field-measured $pCO_2$ in surface water and air, $k$ can be computed by rearranging Equation (3). To compare our calculated $k$ value with other studies, it was standardized to a Schmidt number of 600 ($k_{600}$) by assigning the Schmidt number exponent to be 0.5 (Jähne et al., 1987).

We also predicted the $k_{600}$ (m d$^{-1}$) through the Model 5 developed by Raymond et al. (2012).

$$K_{600} = VS \times 2841 \pm 107 + 2.02 \pm 0.209 \quad (4)$$

where, V is the stream velocity (m s$^{-1}$), S is the slope of rivers (unitless).

Previous studies indicate that $k_{600}$ is affected by a number of environmental factors, such as wind speed,

slope, flow velocity, depth, and discharge (Wanninkhof et al., 1992; Zappa et al., 2007; Raymond et al., 2012). Using only flow velocity and slope of river channels would have caused overestimation for mountainous rivers due to their relatively high channel slope and thus higher flow velocity. Therefore, the extremely high $k_{600}$ values calculated from Equation (3) were excluded from the comparison between our calculated $k_{600}$ and the modeled $k_{600}$.

**3.Results**

**3.1 Characteristics of the hydro-chemical variables**

Water temperature (Tw) varied from 0.1 to 27.7 °C with an average of 11.9±5.7 °C. Average Tw in June (15.1±3.5 °C) and August (17.0±5.4 °C) was considerably higher than that in April (8.4±3.8 °C) and October (7.3±2.4 °C). Seasonal Tw difference was more significant at the wetland (14.4±6.4 °C) and grassland (12.5±5.4 °C) sites than that at the glacier (7.5±4.1 °C) and permafrost (10.0±4.0 °C) sites. Spatial variability of the air temperature was consistent with that of the water temperature at almost all the sites, although it could be as high as 33 °C. The annual average air temperature in 2016 was 16.7±6.3 °C.

Water pH ranged from 6.97 to 9.02 with an average of 7.89±0.64 (Table 1). Mean pH based on all the stream samples was 8.26±0.36, 8.55±0.45, 7.24±0.19, and 7.52±0.36 in April, June, August, and October, respectively. A slight decreasing trend can be observed with the land cover types in the order permafrost > glaciers > grassland > wetland, with the average pH value at 8.13±0.93, 7.93±0.55, 7.85±0.59, and 7.71±0.52, respectively (Table 1). Alkalinity ranged from 600 to 7600 µmol L$^{-1}$ with an average of 2871±1381 µmol L$^{-1}$ (Table 1). Alkalinity was higher in the cold months (3378 µmol L$^{-1}$ in April and 2941 µmol L$^{-1}$ in October) than in the warm months (2644 µmol L$^{-1}$ in June and 2326 µmol L$^{-1}$ in August).

DO values ranged from 2.7 mg L$^{-1}$ to 12.1 mg L$^{-1}$ and the basin-wide mean DO was 7.8±0.6 mg L$^{-1}$ in April, 7.1±1.4 mg L$^{-1}$ in June, 6.7±0.7 mg L$^{-1}$ in August and 7.7±0.7 mg L$^{-1}$ in October, respectively (Table 1). From the perspective of land cover, the highest DO values were observed at the glacier sites, with the annual average at 7.6±0.8 mg L$^{-1}$, followed by the permafrost sites (7.4±1.4 mg L$^{-1}$), the

grassland sites (7.3±0.9 mg L$^{-1}$), and the peatland sites (7.2±1.1 mg L$^{-1}$) (Table 1).

DOC ranged from 0.2 to 12.2 mg L$^{-1}$ with an average of 4.7±2.7 mg L$^{-1}$ (Table 1). DOC exhibited strong seasonality across the rivers. The highest DOC concentration occurred in April (5.0±1.6 mg L$^{-1}$), followed by August (4.9±3.6 mg L$^{-1}$) and June (4.7±2.9 mg L$^{-1}$), and the lowest was found in October (4.0±2.2 mg L$^{-1}$). From the perspective of land cover, the highest DOC concentrations were observed in the peatland with the annual average at 5.1±3.7 mg L$^{-1}$, followed by the permafrost (4.9±2.4mg L$^{-1}$), the grassland (4.6±2.3 mg L$^{-1}$), and the glaciers (3.4±1.1mg L$^{-1}$) (Table 1).

**3.2 Spatial and temporal variations of $p$CO$_2$**

The $p$CO$_2$ ranged from 181 to 2441 µatm with an average of 774±377 µatm, nearly twofold the ambient air $p$CO$_2$. To better illustrate the spatial variability $p$CO$_2$, Figures 2a, 3a, and 2c showed its changes with land cover types. The highest average $p$CO$_2$ value appeared in the peatland (937±4665-6 µatm), followed by grassland (818±394µatm), glacier (645±253 µatm), and the permafrost (600±212 µatm).

The $p$CO$_2$ value showed different temporal variation characteristics for the four land cover types (Figures 2a, 3a, and 2c). In grassland, the average river $p$CO$_2$ value in April, June, August, and October was 836±258 µatm, 609±297 µatm, 1086±551 µatm, and 734±253 µatm, respectively. In comparison, the average peatland river $p$CO$_2$ in April, June, August, and October was 875±436 µatm, 792±436µatm, 1156±630 µatm, and 926±285 µatm, respectively. The $p$CO$_2$ in these two land cover types showed the same temporal pattern with the highest $p$CO$_2$ occurring in August and the lowest in June.

Unlike in the peatland and grassland regions, the riverine $p$CO$_2$ in the glacier and permafrost regions showed relatively small variations but similar seasonal variation trends. In the glacier covered area, the average river $p$CO$_2$ value in April, June, August, and October was 635±122 µatm, 506±31 µatm, 738±449 µatm, and 632±132 µatm respectively. In the permafrost covered area, the average river $p$CO$_2$ value in April, June, August, and October was 465±216 µatm, 586±227 µatm, 591±74 µatm, and 756±231 µatm, respectively.

**3.3 Spatial and temporal variations of $F\mathrm{CO_2}$**

$\mathrm{CO_2}$ emissions exhibited spatial and seasonal variations among the 36 stream sites (Table 1, Figures 2b, 3b, and 3d). The $\mathrm{CO_2}$ effluxes ranged from -221 to 1469 g C m$^{-2}$ yr$^{-1}$ in April, -144 to 6892 g C m$^{-2}$ yr$^{-1}$ in August, and -34 to 2321 g C m$^{-2}$ yr$^{-1}$ in October. While the highest $F\mathrm{CO_2}$ was measured at the wetland sites (Site Pt 3 in August, 6892 g C m$^{-2}$ yr$^{-1}$), the lowest $F\mathrm{CO_2}$ was observed at permafrost sites (Site Pm 3 in April, -221 g C m$^{-2}$ yr$^{-1}$) (Table 1). The averaged $F\mathrm{CO_2}$ of all sites was 479±436, 261±205, 873±1220, and 714±633 g C m$^{-2}$ yr$^{-1}$ in April, June, August, and October, respectively. Clearly, rivers in the Yellow River source region were net carbon sources for the atmosphere, despite the great spatial and seasonal $F\mathrm{CO_2}$ variations. When grouped by land cover types, the mean $\mathrm{CO_2}$ efflux shows a clear decreasing trend from wetland (767±1644 g C m$^{-2}$ yr$^{-1}$) through grassland (679±610 g C m$^{-2}$ yr$^{-1}$) and glacier (508±588 g C m$^{-2}$ yr$^{-1}$) to permafrost (302±±349 g C m$^{-2}$ yr$^{-1}$). Because the intensity of $\mathrm{CO_2}$ emissions depends on river $p\mathrm{CO_2}$, the $F\mathrm{CO_2}$ showed a similar spatial and temporal pattern to the $p\mathrm{CO_2}$, although the highest and lowest $p\mathrm{CO_2}$ and $F\mathrm{CO_2}$ value were not found at the same sampling sites.

**4. Discussion**

**4.1 Impact of land cover types on riverine $p\mathrm{CO_2}$ and $\mathrm{CO_2}$ outgassing**

This study shows that the lowest $F\mathrm{CO_2}$ appeared in the permafrost covered region among all land cover types, with the annual average at $F\mathrm{CO_2}$ of 302±349 g C m$^{-2}$ yr$^{-1}$. It is well known that a large quantity of riverine $\mathrm{CO_2}$ is derived from land (Dinsmore and Billett., 2013; Hope et al., 2004). Particularly, rivers flowing through permafrost are characterized by higher organic carbon input from soils (Zeng et al., 2004), which can support higher riverine DOC export and lead to stronger $\mathrm{CO_2}$ outgassing. The correlation analysis between hydro-chemical parameters and $p\mathrm{CO_2}$ in the permafrost region showed that, while alkalinity, DO and DOC were not significantly correlated with $p\mathrm{CO_2}$, pH exhibited a statistically significant relationship with $p\mathrm{CO_2}$ (Figure 4). The negative relationship between $p\mathrm{CO_2}$ and pH is likely because dissolved $\mathrm{CO_2}$ itself acts as an acid in water (Stumm and Morgan., 1996). In poorly buffered systems like the study area, $\mathrm{CO_2}$ can be a strong control on river water pH (Neal et al., 1998; Waldron et al., 2007). The DOC concentrations in the permafrost rivers (mean: 5.0±2.4 mg L$^{-1}$) were relatively

higher than that in the glacier rivers (mean: 3.6±1.1 mg L$^{-1}$) and the grassland rivers (4.6±2.3 mg L$^{-1}$)

295    but were comparable to the peatland rivers of 5.1±3.7 mg L$^{-1}$ in peatlands. Additionally, the average

alkalinity concentration in the permafrost region is the highest among the four land cover types. However,

the $p$CO$_2$ and $F$CO$_2$ values in this region were always the lowest during the four campaigns. One potential

explanation is that its low temperature (i.e., annual average water temperature: 9.9 °C) because of high

elevation may have constrained soil respiration and riverine organic matter degradation (Battin et al.,

300    2008). Furthermore, although there is sufficient dissolved CO$_2$ in the river water, it may be difficult for

CO$_2$ to degas from rivers in view of the low temperature (thus strong solubility) and low flow velocity

(average: 0.8±0.5 m s$^{-1}$) (Alin et al., 2014). The lower temperature is likely the major reason for the high

riverine DOC concentrations while low CO$_2$ outgassing rates in the permafrost region.

305    Because the glacier region exhibits similar temperatures and elevations to the permafrost, its $p$CO$_2$ and

$F$CO$_2$ values were also relatively low, with the average only at 657±240 g C m$^{-2}$ yr$^{-1}$. This is probably

because all the sampling sites are located on the 1–2 order streams characterized by strong hydrologic

connection with the terrestrial landscape (Sorribas et al., 2017; Smits et al., 2017), and the surrounding

catchment is lack of exogenous terrestrial carbon input. The river water alkalinity of the glacier rivers

310    showed constantly the lowest level throughout the study year (Table 1), due largely to the low coverage

of carbonate rocks. For the glacier rivers, only the DOC was significantly related to $p$CO$_2$ (Figure 5d,

$r^2$=0.56, p < 0.001). The glacier sampling sites are mainly located around the Aemye Ma-chhen Range

(Figure 1). Wang (1998) discovered that these rivers are predominantly supplied by glacier melting that

is characterized by significant seasonal variability. The sampled glacier rivers showed the lowest annual

315    average DOC concentration among the four land cover types (3.6±1.1 mg L$^{-1}$). This is probably because

the sub-catchments around the Aemye Ma-chhen Range do not have sufficient vegetation coverage as a

result of high elevation and low temperature, limiting the terrestrial source of DOC. Poor soil, short water

retention time, and low precipitation are the main reasons for the low vegetation coverage in this region

(Lu et al., 2001). The rivers flowing down the snow mountain cut deep into the B horizon of soils because

320    of strong glacial erosion and retreat. Almost all the glacial sampling sites are characterized by gravel

channel, limiting the supply of terrestrial organic carbon into river carbon pools. As a result, the measured

DOC concentrations in most of the sampled glacier rivers were very low. For glacial rivers, if there is no external supply of DOC, a complete decomposition of the river water DOC can only produce 0.34 μmol $L^{-1}$ $CO_2$. This suggests that the $CO_2$ produced by DOC degradation in the glacial river cannot maintain such a high $CO_2$ outgassing rate. The modern snow and ice which are important water sources in the Aemye Ma-chhen Range do not have enough DOC, DIC, or $CO_2$ contents (Wu et al., 2008). Instead, chemical weathering may have played a crucial role in supporting glacier riverine $CO_2$ (Wu et al., 2005; Wu et al., 2008). Previous studies have shown that glaciers contain large amounts of $CO_2$ (Meese et al., 1997) and DOC (Hood et al., 2009; Singer et al., 2012), which are important sources of $CO_2$ for glacial rivers. Our observations found that, with increasing distance from the glaciers, the riverine $pCO_2$ exhibited a decreasing trend, which is likely caused by the dilution of glacier-related $pCO_2$.

The $FCO_2$ was highest in the peatland rivers among the studied 4 land cover types. Only the pH showed a negative linear relationship with the $pCO_2$, while the alkalinity had a weak linear relationship with the $pCO_2$ (Figure 6). For peatland rivers, terrestrially-derived organic carbon has been widely recognized an important source of riverine $CO_2$ (Abril et al., 2014; Müller et al., 2015; Billett et al., 2015, Hu et al., 2015). There are a variety of sources for DOC in the peatland. First, the soil in the wetland ecosystem is rich in peat soil. The amount of peat stock in the Zoige Peatland is estimated to be 1.9 billion tons, accounting for about 40% of China's marsh wetland carbon storage (Wang et al., 2012). These carbon supplies to river carbon pools are an important driver for the high $FCO_2$ in the wetland rivers. In addition, soil pore water enriched with high concentrations of dissolved $CO_2$ continues to enter river waters, which can provide enough riverine $CO_2$ (Butman et al., 2011). Furthermore, vegetation in the peatland region can import large amounts of $CO_2$ into the river water through two mechanisms. On one hand, vegetation litter and root exudates release degradable organic matter into rivers. Decomposition of these organic matter serves as a carbon source for heterotrophic microorganisms. During this process, heterotrophic organisms release $CO_2$ into water (Abril et al., 2014). On the other hand, respiration of plant roots and soil microorganisms that are submerged in wetland soils could also release $CO_2$ directly into river water (Abril et al., 2014). The combined effects of these factors have resulted in rivers with high DOC and $FCO_2$ values in wetlands.

350

The average $FCO_2$ in the grassland rivers of $818\pm394$ g C m$^{-2}$ yr$^{-1}$ is at a moderate level, lower than the wetland $FCO_2$ but considerably higher than that in the glacier and permafrost rivers. Correlation analyses between water chemistry parameters and riverine $pCO_2$ for the grassland rivers showed that both pH and DOC had weak correlations with $pCO_2$ (Figure 7). This also suggests that $pCO_2$ is partially affected by

355    the water pH. Compared to the other three land cover types, grassland has been substantially affected by human activity (i.e., grazing). Consequently, besides the DOC derived from physical erosion, the pollutants produced by grazing are also important sources of riverine DOC. The average $pCO_2$ in peatland is 15% higher, but the average DOC concentration in wetland is 11% higher than that in grassland, and the alkalinity in grassland rivers is 46% higher than that in the wetland rivers. In addition, DIC is an

360    important source of riverine $CO_2$ for grassland rivers. While stream DIC source are highly variable across space and time (Smits et al., 2017), most of the HCO$_3^-$ in the Yellow River source region is derived from carbonate and silicate weathering (Wu et al., 2005; Wu et al., 2008; Wu et al., 2008), which largely reflects the contribution of groundwater inflow (Marx et al., 2017). Our groundwater samples from grassland region show an average $pCO_2$ of 1976 μatm, which is 2.5 times the average $pCO_2$ of the whole

365    Yellow River source region. Therefore, the $CO_2$ excess in the grassland rivers is more likely maintained by both the terrestrial organic carbon input and the inorganic carbon from groundwater.

With respect to the $k_{600}$, the computed $k_{600}$ showed statistically significant but weak correlation with the modeled results (Figure 8a) when the high $k_{600}$ values ($>70$ m d$^{-1}$) were removed from analysis. Given

370    the chamber's dampening effect of wind (Matthews et al., 2003), there was no any statistically significant relationship between wind and $k_{600}$ for streams. Instead, flow velocity is a relatively good predictor of $k_{600}$ and can approximately explain 15% of its variability (Figure 8b). Although we deployed the floating chamber very carefully, the statistical analysis could not reflect the complex interactions of various environment factors except the four land cover types through our 36 sampling sites. Additionally, it is

375    worth noting that the Model 5 of Raymond et al. (2012) has overestimated the $k_{600}$, especially for mountainous rivers. This is probably because of low water temperature that has constrained $CO_2$ degassing although the steeper channel slope has caused stronger flow turbulence (Battin et al., 2008). A

low temperature will limit the rate of Brownian motion and reduce the $CO_2$ exchange with the atmosphere. Meanwhile, a low temperature will increase the solubility of dissolved $CO_2$, thus reducing the outgassing of $CO_2$.

**4.2 Significance and implications for riverine carbon budgets**

This study demonstrates that the annual average $pCO_2$ is 771±380 µatm and $FCO_2$ is 590±766 g C m$^{-2}$ yr$^{-1}$ in the Yellow River source region. In comparison, Ran et al. (2015a; 2015b) estimated a considerably lower $pCO_2$ value of 241±79 µatm and an areal $CO_2$ efflux of is -221±112 g C m$^2$ yr$^{-1}$ for the Yellow River source region, indicative of a strong carbon uptake from the atmosphere. Combining the seasonal difference of water surface area between the wet season (122 days and a water surface area of 770 km$^2$) and the dry season (243 days and a water surface area of 560 km$^2$), we estimated a total $CO_2$ efflux from the Yellow River source region at 0.37±0.49 T g C yr$^{-1}$. This suggests a net carbon source for the atmosphere. Our $CO_2$ effluxes contrast with the earlier estimate by Ran et al. (2015b) which reported a carbon sink of -0.17±0.08 Tg C yr$^{-1}$.

Unlike our systematic sampling within the Yellow River source region, Ran et al. (2015b) estimated its riverine $CO_2$ outgassing by using only results at five sampling sites. There may have caused the huge $CO_2$ efflux difference. Firstly, the sampling by Ran et al. (2015b) was confined to the mainstem and major tributaries, which may have underestimated $CO_2$ emissions from lower-order headwater streams that usually present strong $CO_2$ degassing (Butman and Raymond, 2011). For example, our sampling in the Zoige peatland rivers demonstrated that the lower-order rivers exhibit substantially higher $FCO_2$ (767±1144 g C m$^{-2}$ yr$^{-1}$) than the Yellow River mainstem (351±306 g C m$^{-2}$ yr$^{-1}$). This reveals the impact of strong flow turbulence and land-river connectivity of low-order streams on sustaining the high $CO_2$ effluxes (Crawford et al., 2013). In addition, the importance of groundwater inflow may decline with increasing stream orders, leading to a decreasing $pCO_2$ and thus lower $FCO_2$ (Marx et al., 2017). Another potential reason is that the number of sampling sites has limited the accuracy of $CO_2$ emissions. This is highly possible for the Yellow River source region with the $pCO_2$ in groundwater (1976 µatm) 2.5 times higher than that in the river (771±380 µatm). The $CO_2$ originating from groundwater can be quickly

released to the atmosphere within a short distance (Hotchkiss et al., 2015). Obviously, it is considerably challenging to detect the impact of groundwater inflow without high-resolution sampling.

While the Yellow River source region occupies 17.6% of the whole Yellow River basin, it accounts for only around 4% of the basin's total $CO_2$ efflux (Ran et al., 2015a; 2015b). The $CO_2$ efflux of the Yellow River source region is also small compared with the effluxes from boreal river catchments (Teodoru et al., 2009; Butman and Raymond., 2011; Crawford et al., 2013; 2015; Kokic et al., 2015; Looman et al., 2016) or even smaller relative to the global $CO_2$ efflux (Aufdenkampe et al., 2011). Nevertheless, there is a huge carbon emission potential in the coming decades. Since the permafrost and wetland in the Yellow River source region are abundant in huge quantities of carbon storage. Continuously increasing temperature due to global warming will accelerate not only the mobilization of organic carbon in permafrost, but also the degradation of organic carbon by soil microorganisms. As a consequence, increasing riverine $CO_2$ effluxes are highly anticipated and warrant further studies to comprehensively understand their implications for global carbon cycle and climate change.

We have comprehensively evaluated the riverine carbon dynamics within the Yellow River source region by means of *in situ* measurement of $CO_2$ emissions under four different land cover types. However, it must be noted that there are still great uncertainties to be properly addressed in future studies. Despite the significant increase in the number of sampling sites compared with previous studies, less research on single watersheds that are spatially representative has been performed. Moreover, temporally continuous sampling involving the diel dynamics of riverine carbon export remains lacking. For example, prior studies suggest $CO_2$ efflux during the daytime would be completely different from that at night and floods may have a huge shift on $CO_2$ emissions (Geeraert et al., 2017; Smits et al., 2017).

**5. Conclusions**

Based on four rounds of field direct measurements of $CO_2$ outgassing within the Yellow River source region, the average $pCO_2$ in the study area was estimated at $771\pm380\,\mu atm$ and the average $FCO_2$ was $590\pm766\,g\,C\,m^{-2}\,yr^{-1}$. The $FCO_2$ and $pCO_2$ are lower than other rivers in the world and at a relatively

low level compared to the middle and lower reaches of the Yellow River. The results showed that the rivers in the Yellow River source region were net sources of atmospheric $CO_2$. Both the $pCO_2$ and $FCO_2$ showed strong spatial and temporal variations. The largest riverine $CO_2$ efflux was found in August, followed by October and April, and the lowest was observed in June. When grouped into different land cover types, the $FCO_2$ in the permafrost river was the lowest among the four types of land cover. The highest $FCO_2$ was found in peatland rivers, followed by rivers in the grassland and glacier regions.

For the Yellow River source region with an alpine climate, the low temperature conditions have played a crucial role in limiting its biological activity and reducing $CO_2$ emissions. As a consequence, these procedures control both the riverine $CO_2$ sources and gas transfer velocity across the water-air interface. The DOC concentration acts as an important control on riverine $CO_2$ dynamics under all the four land cover types. In the permafrost region, the large amounts of terrestrially-derived DOC supported its high $pCO_2$ levels. While in the glacier region, the glacial DOC and $CO_2$ may have played an essential role in determining $CO_2$ outgassing. In the peatland and grassland regions, decomposition of plant-derived organic matter is an important source of riverine $CO_2$. Moreover, groundwater inflow and chemical weathering played an important role in supporting riverine $CO_2$ for the whole Yellow River source region.

By integrating the seasonal changes of water surface area, the riverine $CO_2$ efflux of the Yellow River source region was estimated at $0.37\pm0.49$ Tg C yr$^{-1}$, which is significantly higher than earlier estimates (e.g., $-0.168\pm0.084$ Tg C yr$^{-1}$ by Ran et al. (2015a; 2015b). To date, very few studies have focused on the dynamics of riverine carbon cycling on the Tibetan Plateau river systems. This study provides insight into the riverine $CO_2$ outgassing in the Yellow River source region, which will improve our current understanding of $CO_2$ emissions from alpine rivers in the world, in particular these located on the Tibetan Plateau.

**Acknowledgements:** This research was supported by the Natural Science Foundation of China (Grant No. 91547110; 51469018) and the National University of Singapore (Grant No. R-109-000-191-646; R-109-000-227-115). Special thanks go to the two anonymous reviewers for their constructive comments

which greatly improved the manuscript.

[revised manuscript text omitted]

region in 2016.

[Figure]

Figure 3. The box plots of $pCO_2$ and $FCO_2$ under four different land cover types within the Yellow River source region, expressed in the order of April, June, August, and October of 2016 (a and b). The $pCO_2$ data expressed in the order of grassland, peatland, glacier, permafrost, and groundwater (c), The $FCO_2$ expressed in the order of grassland, peatland, glacier, and groundwater (d).

[Figure]

Figure 4. The linear relationship of hydro-chemical parameters and $p\mathrm{CO_2}$ in permafrost covered region. (a) pH, (b) alkalinity, (c) dissolved oxygen, and (d) dissolved organic carbon.

[Figure]

Figure 5. The linear relationship of hydro-chemical parameters and $p\mathrm{CO_2}$ in glacier covered region. (a) pH, (b) alkalinity, (c) dissolved oxygen, and (d) dissolved organic carbon.

[Figure]

Figure 6. The linear relationship of hydro-chemical parameters and $p$CO$_2$ in peatland covered region. (a) pH, (b) alkalinity, (c) dissolved oxygen, and (d) dissolved organic carbon.

[Figure]

Figure 7. The linear relationship of hydro-chemical parameters and $p$CO$_2$ in grassland covered region. (a) pH, (b) alkalinity, (c) dissolved oxygen, and (d) dissolved organic carbon.

[Figure]

Figure 8. (a) The relationship between actual measurements (based on *in situ* $p$CO$_2$ and $F$CO$_2$) and predicted $k_{600}$ using the Model 5 of Raymond et al. (2012) for streams; (b) Correlation between standardized (based on *in situ* $p$CO$_2$ and $F$CO$_2$) gas transfer velocity ($k_{600}$) and flow velocity over the 4 campaigns of field sampling.